# Benign overfitting in leaky ReLU networks with moderate input dimension

**Kedar Karhadkar**[1*]  **Erin George**[1*]  **Michael Murray**[1]  **Guido Montúfar**[12]  **Deanna Needell**[1]

{kedar,egeo,mmurray,montufar,deanna}@math.ucla.edu
[1]UCLA    [2]Max Planck Institute for Mathematics in the Sciences
[*]Equal contribution

## Abstract

The problem of benign overfitting asks whether it is possible for a model to perfectly fit noisy training data and still generalize well. We study benign overfitting in two-layer leaky ReLU networks trained with the hinge loss on a binary classification task. We consider input data that can be decomposed into the sum of a common signal and a random noise component, that lie on subspaces orthogonal to one another. We characterize conditions on the signal to noise ratio (SNR) of the model parameters giving rise to benign versus non-benign (or harmful) overfitting: in particular, if the SNR is high then benign overfitting occurs, conversely if the SNR is low then harmful overfitting occurs. We attribute both benign and non-benign overfitting to an approximate margin maximization property and show that leaky ReLU networks trained on hinge loss with gradient descent (GD) satisfy this property. In contrast to prior work we do not require the training data to be nearly orthogonal. Notably, for input dimension $d$ and training sample size $n$, while results in prior work require $d = \Omega(n^2 \log n)$, here we require only $d = \Omega(n)$.

## 1 Introduction

Intuition from learning theory suggests that fitting noise during training reduces a model's performance on test data. However, it has been observed in some settings that machine learning models can interpolate noisy training data with only *nominal* cost to their generalization performance (Zhang et al., 2017; Belkin et al., 2018, 2019), a phenomenon referred to as *benign overfitting*. Establishing theory that can explain this phenomenon has attracted much interest in recent years and there is now a rich body of work on this topic particularly in the context of linear models. However, the study of benign overfitting in the context of non-linear models, in particular shallow ReLU or leaky ReLU networks, has additional technical challenges and subsequently is less well advanced.

Much of the effort in regard to theoretically characterizing benign overfitting focuses on showing, under an appropriate scaling of the dimension of the input domain $d$, size of the training sample $n$, number of corruptions $k$ and number of model parameters $p$ that a model can interpolate noisy training data while achieving an arbitrarily small generalization error. Such characterizations of benign overfitting position it as a *high dimensional phenomenon*[1]: indeed, the decrease in generalization error is achieved by escaping to higher dimensions at some rate relative to the other aforementioned hyperparameters. However, for these mathematical results to be relevant for explaining benign overfitting as observed in practice, clearly the particular scaling of $d$ with respect to $n, k$ and $p$ needs to reflect the ratios seen in practice. Although a number of works, which we discuss in Section 1.2, establish benign overfitting results for shallow neural networks, a key and significant limitation they share is the requirement that the input features of the training data are at least approximately

---

[1]We provide a formal definition of benign overfitting as a high dimensional phenomenon in Appendix E.

38th Conference on Neural Information Processing Systems (NeurIPS 2024).

orthogonal to one another. To study benign overfitting, these prior works typically assume the input features consist of a small, low-rank signal component plus an isotropic noise term. Therefore, for the near orthogonality property to hold with high probability it is required that the input dimension $d$ scales as $d = \Omega(n^2 \log n)$ or higher. This assumption highly restricts the applicability of these results for explaining benign overfitting in practice.

In this work we assume only $d = \Omega(n)$ and establish both harmful and benign overfitting results for shallow leaky ReLU networks trained via gradient descent (GD) on the hinge loss. In particular, we consider $n$ data point pairs $(\boldsymbol{x}_i, y_i) \in \mathbb{R}^d \times \{\pm 1\}$, where, for some vector $\boldsymbol{v} \in \mathbb{S}^{d-1}$ and scalar $\gamma \in [0, 1]$, the input features are drawn from a pair of Gaussian clusters $\boldsymbol{x}_i \sim \mathcal{N}(\pm\sqrt{\gamma}\boldsymbol{v}, \sqrt{1-\gamma}\frac{1}{d}(\mathbf{I}_d - \boldsymbol{v}\boldsymbol{v}^T))$ and $y_i = \operatorname{sign}(\langle \mathbb{E}[\boldsymbol{x}_i], \boldsymbol{v} \rangle)$. The training data is noisy in that $k$ of the $n$ points in the training sample have their output label flipped. We assume equal numbers of positive and negative points among clean and corrupt ones. We provide a full description of our setup and assumptions in Section 2. Our proof techniques are novel and identify a new condition allowing for the analysis of benign and harmful overfitting which we term *approximate margin maximization*, wherein the norm of the network parameters is upper bounded by a constant of the norm of the max-margin linear classifier.

## 1.1 Summary of contributions

Our key results are summarized as follows.

- In Theorem 3.1, we prove that a leaky ReLU network trained on linearly separable data with gradient descent and the hinge loss will attain zero training loss in finitely many iterations. Moreover, the network weight matrix $\boldsymbol{W}$ at convergence will be approximately max-margin in the sense that $\|\boldsymbol{W}\| = O\left(\frac{\|\boldsymbol{w}^*\|}{\alpha\sqrt{m}}\right)$, where $\alpha$ is the leaky parameter of the activation function, $m$ is the width of the network, and $\boldsymbol{w}^*$ is the max-margin linear classifier. We apply this result to derive generalization bounds for the network on test data.

- In Theorem 3.2, we establish conditions under which benign overfitting occurs for leaky ReLU networks. If the input dimension $d$, number of training points $n$, number of corrupt points $k$, and signal strength $\gamma$ satisfy $d = \Omega(n)$ and $\gamma = \Omega(\frac{1}{k})$, then the network will exhibit benign overfitting. We emphasize that existing works on benign overfitting require $d = \Omega(n^2 \log n)$ to ensure nearly orthogonal data.

- In Theorem 3.3, we find a complementary lower bound for the generalization error to show that, for gradient descent classifiers, the bound in Theorem 3.2 is tight up to a constant in the exponent that can depend on $\alpha$.

- In Theorem 3.4, we find conditions under which non-benign overfitting occurs. If $d = \Omega(n)$ and $\gamma = O(\frac{1}{d})$, then the network will exhibit non-benign overfitting: in particular its generalization error will be at least $\frac{1}{8}$.

## 1.2 Related work

There is now a significant body of literature theoretically characterizing benign overfitting in the context of linear models, including linear regression (Bartlett et al., 2020; Muthukumar et al., 2020; Wu & Xu, 2020; Zou et al., 2021; Hastie et al., 2022; Koehler et al., 2021; Wang et al., 2021a; Chatterji & Long, 2022; Shamir, 2022), logistic regression (Chatterji & Long, 2021; Muthukumar et al., 2021; Wang et al., 2021b), max-margin classification with linear and random feature models (Montanari et al., 2023b,a; Mei & Montanari, 2022; Cao et al., 2021) and kernel regression (Liang & Rakhlin, 2020; Liang et al., 2020; Adlam & Pennington, 2020). However, the study of benign overfitting in non-linear models is more nascent.

Homogeneous networks trained with gradient descent and an exponentially tailed loss are known to converge in direction to a Karush-Kuhn-Tucker (KKT) point of the associated max-margin problem (Lyu & Li, 2020; Ji & Telgarsky, 2020)[2]. This property has been widely used in prior works to prove benign overfitting results for shallow neural networks. Frei et al. (2022) consider a shallow, smooth leaky ReLU network trained with an exponentially tailed loss and assume the data is drawn from a mixture of well-separated sub-Gaussian distributions. A key result of this work is, given

---

[2]One also needs to assume initialization from a position with a low initial loss.

sufficient iterations of GD, that the network will interpolate noisy training data while also achieving minimax optimal generalization error up to constants in the exponents. Xu & Gu (2023) extend this result to more general activation functions, including ReLU, as well as relax the assumptions on the noise distribution to being centered with bounded logarithmic Sobolev constant, and finally also improve the convergence rate. George et al. (2023) also study ReLU as opposed to leaky ReLU networks but do so in the context of the hinge loss, for which, and unlike exponentially tailed losses, a characterization of the implicit bias is not known. This work also establishes transitions on the margin of the clean data driving harmful, benign and no-overfitting training outcomes. Frei et al. (2023) use the aforementioned implicit bias of GD for linear classifiers and shallow leaky ReLU networks towards solutions that satisfy the KKT conditions of the margin maximization problem to establish settings where the satisfaction of said KKT conditions implies benign overfitting. Kornowski et al. (2023) also use the implicit bias results for exponentially tailed losses to derive similar benign overfitting results for shallow ReLU networks. Cao et al. (2022); Kou et al. (2023) study benign overfitting in two-layer convolutional as opposed to feedforward neural networks: indeed, whereas in most prior works data is modeled as the sum of a signal and noise component, in these two works the signal and noise components are assumed to lie in disjoint patches. The weight vector of each neuron is applied to both patches separately and a non-linearity, such as ReLU, is applied to the resulting pre-activation. In this setting, the authors prove interpolation of the noisy training data and derive conditions on the clean margin under which the network benignly versus harmfully overfits. A follow up work (Chen et al., 2023) considers the impact of Sharpness Aware Minimization (SAM) in the same setting. Finally, and assuming $d = \Omega(n^5)$, Xu et al. (2024) establish benign overfitting results for a data distribution which, instead of being linearly separable, is separated according to an XOR function.

We emphasize that the prior work on benign overfitting in the context of shallow neural networks requires the input data to be approximately orthogonal. Under standard data models studied this equates to the requirement that the input dimension $d$ versus the size of the training sample $n$ satisfies $d = \Omega(n^2 \log n)$ or higher. Here we require only $d = \Omega(n)$. The weaker dimensionality requirement requires substantially different proof techniques. George et al. (2023) study a setting most similar to the one studied here, however, the techniques are very different. In particular, the results presented in this other work are derived by carefully tracking neuron activation patterns. While in high dimensions this is feasible due to the near orthogonality of the noise in low dimensions this is far more challenging as noise vectors can be highly correlated leading to coupling effects.

Finally, we remark that our proof technique for the convergence of GD to a global minimizer in the context of a shallow leaky ReLU network (Theorem 3.1) is closely related to the proof techniques used by Brutzkus et al. (2018). While this work does establish a generalization bound, the bound assumes that population dataset is linearly separable rather than just the training dataset. Hence, it cannot be applied when the training dataset has label-flipping noise, which is the setting that we are interested in for benign overfitting.

## 2 Preliminaries

Let $[n] = \{1, 2, \ldots, n\}$ denote the set of the first $n$ natural numbers. We remark that when using big-$O$ notation we implicitly assume only positive constants. We use $c, C, C_1, C_2, \ldots$ to denote absolute constants with respect to the input dimension $d$, the training sample size $n$, and the width of the network $m$. Note constants may change in value from line to line. Furthermore, when using big-$O$ notation all variables aside from $d, n, k$ and $m$ are considered constants. However, for clarity we will frequently make the constants concerning the confidence $\delta$ and failure probability $\epsilon$ explicit. Moreover, for two functions $f, g : \mathbb{N} \to \mathbb{N}$, if we say $f = O(g)$ implies property $p$, what we mean is there exists an $N \in \mathbb{N}$ and a constant $C$ such that if $f(n) \leq Cg(n)$ for all $n \geq N$ then property $p$ holds. Likewise, if we say $f = \Omega(g)$ implies property $p$, what we mean is there exists an $N \in \mathbb{N}$ and a constant $C$ such that if $f(n) \geq Cg(n)$ for all $n \geq N$ then property $p$ holds. Finally, we use $\|\cdot\|$ to denote the $\ell^2$ norm of the vector argument or $\ell^2 \to \ell^2$ operator norm of the matrix argument.

### 2.1 Data model

We study data generated as per the following data model.

**Definition 2.1.** *Suppose $d, n, k \in \mathbb{N}$, $\gamma \in (0,1)$ and $\boldsymbol{v} \in \mathbb{S}^{d-1}$. If $(\boldsymbol{X}, \hat{\boldsymbol{y}}, \boldsymbol{y}, \boldsymbol{x}, y) \sim \mathcal{D}(d, n, k, \gamma, \boldsymbol{v})$ then*

1. *$\boldsymbol{X} \in \mathbb{R}^{n \times d}$ is a random matrix whose rows, which we denote $\boldsymbol{x}_i$, satisfy $\boldsymbol{x}_i = \sqrt{\gamma} y_i \boldsymbol{v} + \sqrt{1 - \gamma} \boldsymbol{n}_i$, where $\boldsymbol{n}_i \sim \mathcal{N}(\boldsymbol{0}_d, \frac{1}{d}(\boldsymbol{I}_d - \boldsymbol{v}\boldsymbol{v}^T))$ are mutually i.i.d..*

2. *$\boldsymbol{y} \in \{\pm 1\}^n$ is a random vector with entries $y_i$ that are mutually independent of one another as well as the noise vectors $(\boldsymbol{n}_i)_{i \in [n]}$ and are uniformly distributed over $\{\pm 1\}$. This vector holds the true labels of the training set.*

3. *Let $\mathcal{B} \subset [n]$ be any subset chosen independently of $\boldsymbol{y}$ such that $|\mathcal{B}| = k$. Then $\hat{\boldsymbol{y}} \in \{\pm 1\}^n$ is a random vector whose entries satisfy $\hat{y}_i \neq y_i$ for all $i \in \mathcal{B}$ and $\hat{y}_i = y_i$ for all $i \in \mathcal{B}^c =: \mathcal{G}$. This vector holds the observed labels of the training set.*

4. *$y$ is a random variable representing a test label which is uniformly distributed over $\{\pm 1\}$.*

5. *$\boldsymbol{x} \in \mathbb{R}^d$ is a random vector representing the input feature of a test point and satisfies $\boldsymbol{x} = \sqrt{\gamma} y \boldsymbol{v} + \sqrt{1 - \gamma} \boldsymbol{n}$, where $\boldsymbol{n} \sim \mathcal{N}(\boldsymbol{0}_d, \frac{1}{d}(\boldsymbol{I}_d - \boldsymbol{v}\boldsymbol{v}^T))$ is mutually independent of the random vectors $(\boldsymbol{n}_i)_{i \in [n]}$.*

*We refer to $(\boldsymbol{X}, \hat{\boldsymbol{y}})$ as the training data and $(\boldsymbol{x}, y)$ as the test data. Furthermore, for typographical convenience we define $\boldsymbol{y} \odot \hat{\boldsymbol{y}} =: \beta \in \{\pm 1\}^n$.*

To provide some interpretation to Definition 2.1, the training data consists of $n$ points of which $k$ have their observed label flipped relative to the true label. We refer to $\boldsymbol{v}$ and $\boldsymbol{n}_i$ as the signal and noise components of the $i$-th data point respectively: indeed, with $\gamma > 0$ then for $i \in \mathcal{G}$ $y_i \langle \boldsymbol{x}_i, \boldsymbol{v} \rangle = \sqrt{\gamma} > 0$. The test data is drawn from the same distribution and is assumed not to be corrupted. We say that the training data $(\boldsymbol{X}, \hat{\boldsymbol{y}})$ is *linearly separable* if there exists $\boldsymbol{w} \in \mathbb{R}^d$ such that

$$\hat{y}_i \langle \boldsymbol{w}, \boldsymbol{x}_i \rangle \geq 1, \quad \text{for all } i \in [n].$$

For finite $n$, this condition is equivalent to the existence of a $\boldsymbol{w}$ with $\hat{y}_i \langle \boldsymbol{w}, \boldsymbol{x}_i \rangle > 0$ for all $i \in [n]$. We denote the set of linearly separable datasets as $\mathcal{X}_{lin} \subset \mathbb{R}^{n \times d} \times \{\pm 1\}^n$. For a linearly separable dataset $(\boldsymbol{X}, \hat{\boldsymbol{y}})$, the *max-margin linear classifier* is the unique solution to the optimization problem

$$\arg\min_{\boldsymbol{w} \in \mathbb{R}^d} \|\boldsymbol{w}\| \text{ such that } \hat{y}_i \langle \boldsymbol{w}, \boldsymbol{x}_i \rangle \geq 1 \text{ for all } i \in [n].$$

Observe one may equivalently take a strictly convex objective $\|\boldsymbol{w}\|^2$ and the constraint set is a closed convex polyhedron that is non-empty iff the data is linearly separable. The max-margin linear classifier $\boldsymbol{w}^*$ has a corresponding geometric margin $2/\|\boldsymbol{w}^*\|$. When $d \geq n$ and $\gamma > 0$, input feature matrices $\boldsymbol{X}$ from our data model almost surely have linearly independent rows $\boldsymbol{x}_i$ and thus $(\boldsymbol{X}, \hat{\boldsymbol{y}})$ is almost surely linearly separable for any observed labels $\hat{\boldsymbol{y}} \in \{\pm 1\}^n$.

## 2.2 Architecture and learning algorithm

We study shallow leaky ReLU networks with a forward pass function $f : \mathbb{R}^{2m \times d} \times \mathbb{R}^d \to \mathbb{R}$ defined as

$$f(\boldsymbol{W}, \boldsymbol{x}) = \sum_{j=1}^{2m} (-1)^j \sigma(\langle \boldsymbol{w}_j, \boldsymbol{x} \rangle), \tag{1}$$

where $\boldsymbol{W} \in \mathbb{R}^{2m \times d}$ are the parameters of the network, $\sigma : \mathbb{R} \to \mathbb{R}$ is the leaky ReLU function, defined as $\sigma(x) = \max(x, \alpha x)$, where $\alpha \in (0, 1]$ is referred to as the leaky parameter. We remark that we only train the weights of the first layer and keep the output weights of each neuron fixed. Although $\sigma$ is not differentiable at 0, in the context of gradient descent we adopt a subgradient and let $\dot{\sigma}(z) = 1$ for $z \geq 0$ and let $\dot{\sigma}(z) = \alpha$ otherwise. The *hinge loss* $\ell : \mathbb{R} \to \mathbb{R}_{\geq 0}$ is defined as

$$\ell(z) = \max\{0, 1 - z\}. \tag{2}$$

Again, $\ell$ is not differentiable at zero; adopting a subgradient we define for any $j \in [2m]$

$$\nabla_{\boldsymbol{w}_j} \ell(\hat{y} f(\boldsymbol{W}, \boldsymbol{x})) = \begin{cases} (-1)^{j+1} \hat{y} \boldsymbol{x} \dot{\sigma}(\langle \boldsymbol{w}_j, \boldsymbol{x} \rangle) & \hat{y} f(\boldsymbol{W}, \boldsymbol{x}) < 1, \\ 0 & \hat{y} f(\boldsymbol{W}, \boldsymbol{x}) \geq 1. \end{cases}$$

The training loss $L : \mathbb{R}^{2m \times d} \times \mathbb{R}^{n \times d} \times \mathbb{R}^n \to \mathbb{R}$ is defined as

$$L(\boldsymbol{W}, \boldsymbol{X}, \hat{\boldsymbol{y}}) = \sum_{i=1}^n \ell(\hat{y}_i f(\boldsymbol{W}, \boldsymbol{x}_i)). \tag{3}$$

Let $\boldsymbol{W}^{(0)} \in \mathbb{R}^{2m \times d}$ denote the model parameters at initialization. For each $t \in \mathbb{N}$ we define $\boldsymbol{W}^{(t)}$ recursively as

$$\boldsymbol{W}^{(t)} = \boldsymbol{W}^{(t-1)} - \eta \nabla_{\boldsymbol{W}} L(\boldsymbol{W}^{(t-1)}, \boldsymbol{X}, \hat{\boldsymbol{y}}),$$

where $\eta > 0$ is the step size. Let $\mathcal{F}^{(t)} \subseteq [n]$ denote the set of all $i \in [n]$ such that $\hat{y}_i f(\boldsymbol{W}^{(t)}, \boldsymbol{x}_i) < 1$. Then equivalently each neuron is updated according to the following rule: for $j \in [2m]$

$$\boldsymbol{w}_j^{(t)} = GD(\boldsymbol{W}^{(t-1)}, \eta) := \boldsymbol{w}_j^{(t-1)} + \eta(-1)^j \sum_{i \in \mathcal{F}^{(t-1)}} \hat{y}_i \boldsymbol{x}_i \dot{\sigma}(\langle \boldsymbol{w}_j^{(t-1)}, \boldsymbol{x}_i \rangle). \tag{4}$$

For ease of reference we now provide the following definition of the learning algorithm described above.

**Definition 2.2.** *Let $\mathcal{A}_{GD} : \mathbb{R}^{n \times d} \times \{\pm 1\}^n \times \mathbb{R} \times \mathbb{R}^{2m \times d} \to \mathbb{R}^{2m \times d}$ return $\mathcal{A}_{GD}(\boldsymbol{X}, \hat{\boldsymbol{y}}, \eta, \boldsymbol{W}^{(0)}) =: \boldsymbol{W}$, where the $j$-th row $\boldsymbol{w}_j$ of $\boldsymbol{W}$ is defined as follows: let $\boldsymbol{w}_j^{(0)}$ be the $j$-th row of $\boldsymbol{W}^{(0)}$ and generate the sequence $(\boldsymbol{w}_j^{(t)})_{t \geq 0}$ using the recurrence relation $\boldsymbol{w}_j^{(t)} = GD(\boldsymbol{W}^{(t-1)}, \eta)$ as defined in equation 4.*

1. *If for $j \in [2m]$, $\lim_{t \to \infty} \boldsymbol{w}_j^{(t)}$ does not exist then we say $\mathcal{A}_{GD}$ is undefined.*

2. *Otherwise we say $\mathcal{A}_{GD}$ converges and $\boldsymbol{w}_j = \lim_{t \to \infty} \boldsymbol{w}_j^{(t)}$.*

3. *If there exists a $T \in \mathbb{N}$ such that for all $j \in [2m]$ $\boldsymbol{w}_j^{(t)} = \boldsymbol{w}_j^{(T)}$ for all $t \geq T$, then we say $\mathcal{A}_{GD}$ converges in finite time.*

We often find that all matrices in the set

$$\{\mathcal{A}_{GD}(\boldsymbol{X}, \hat{\boldsymbol{y}}, \eta, \boldsymbol{W}^{(0)}) : \forall j \, \|\boldsymbol{w}_j^{(0)}\| \leq \lambda\}$$

agree on all relevant properties. In this case, we abuse notation and say that $\mathcal{A}_{GD}(\boldsymbol{X}, \hat{\boldsymbol{y}}, \eta, \lambda) = \boldsymbol{W}$ where $\boldsymbol{W}$ is a generic element from this set.

Finally, in order to derive our results we make the following assumptions concerning the step size and initialization of the network.

**Assumption 1.** *The step size $\eta$ satisfies $\eta \leq 1/(mn \max_{i \in [n]} \|\boldsymbol{x}_i\|^2)$ and for all $j \in [2m]$ the network at initialization satisfies $\|\boldsymbol{w}_j^{(0)}\| \leq \sqrt{\alpha}/(m \min_{i \in [n]} \|\boldsymbol{x}_i\|)$.*

Under our data model the input data points have approximately unit norm; therefore these assumptions reduce to $\eta \leq \frac{C}{mn}$ and $\|\boldsymbol{w}_j^{(0)}\| \leq \frac{C\sqrt{\alpha}}{m}$.

## 2.3 Approximate margin maximization

We now introduce the notion of an approximate margin maximizing algorithm, which plays a key role in deriving our results. Although the primary setting we consider in this work is the learning algorithm $\mathcal{A}_{GD}$ (see Definition 2.2), we derive benign overfitting guarantees more broadly for any learning algorithm which fits into this category. Recall $\mathcal{X}_{lin}$ denotes the set of linearly separable datasets $(\boldsymbol{X}, \hat{\boldsymbol{y}}) \in \mathbb{R}^{n \times d} \times \{\pm 1\}^n$.

**Definition 2.3.** *Let $f : \mathbb{R}^p \times \mathbb{R}^d \to \mathbb{R}$ denote a predictor function with $p$ parameters. An algorithm $\mathcal{A} : \mathbb{R}^{n \times d} \times \mathbb{R}^n \to \mathbb{R}^p$ is approximately margin maximizing with factor $M > 0$ on $f$ if for all $(\boldsymbol{X}, \hat{\boldsymbol{y}}) \in \mathcal{X}_{lin}$*

$$\hat{y}_i f(\mathcal{A}(\boldsymbol{X}, \hat{\boldsymbol{y}}), \boldsymbol{x}_i) \geq 1 \text{ for all } i \in [n] \tag{5}$$

*and*

$$\|\mathcal{A}(\boldsymbol{X}, \hat{\boldsymbol{y}})\| \leq M \|\boldsymbol{w}^*\|, \tag{6}$$

*where $\boldsymbol{w}^*$ is the max-margin linear classifier of $(\boldsymbol{X}, \hat{\boldsymbol{y}})$. Moreover, if $\mathcal{A}$ is an approximate margin maximizing algorithm we define*

$$|\mathcal{A}| = \inf\{M > 0 : \mathcal{A} \text{ is approximately margin maximizing with factor } M\}. \tag{7}$$

In the above definition we take the standard Euclidean norm on $\mathbb{R}^p$. In particular if $\mathbb{R}^p = \mathbb{R}^{2m \times d}$ is a space of matrices we take the Frobenius norm.

# 3   Main results

In order to prove benign overfitting it is necessary to show that the learning algorithm outputs a model that correctly classifies all points in the training sample. The following theorem establishes this for $\mathcal{A}_{GD}$ and bounds the margin maximizing factor $|\mathcal{A}_{GD}|$.

**Theorem 3.1.** *Let $f : \mathbb{R}^p \times \mathbb{R}^n \to \mathbb{R}$ be a leaky ReLU network with forward pass as defined by equation 1. Suppose the step size $\eta$ and initialization condition $\lambda$ satisfy Assumption 1. Then for any linearly separable data set $(\boldsymbol{X}, \hat{\boldsymbol{y}})$ $\mathcal{A}_{GD}(\boldsymbol{X}, \hat{\boldsymbol{y}}, \eta, \lambda)$ converges after $T$ iterations, where*

$$T \le \frac{C\|\boldsymbol{w}^*\|^2}{\eta \alpha^2 m}.$$

*Furthermore $\mathcal{A}_{GD}$ is approximately margin maximizing on $f$ (Definition 2.3) with*

$$|\mathcal{A}_{GD}| \le \frac{C}{\alpha\sqrt{m}}.$$

A proof of Theorem 3.1 can be found in Appendix D.1. Note also by Definition 2.3 that the solution $\boldsymbol{W} = \mathcal{A}_{GD}(\boldsymbol{X}, \hat{\boldsymbol{y}})$ for $(\boldsymbol{X}, \hat{\boldsymbol{y}}) \in \mathcal{X}_{lin}$ is a global minimizer of the training loss defined in equation 3 with $L(\boldsymbol{W}, \boldsymbol{X}, \hat{\boldsymbol{y}}) = 0$. Our approach to proving this result is reminiscent of the proof of convergence of the perceptron algorithm and therefore is also similar to the techniques used by Brutzkus et al. (2018).

For training and test data as per Definition 2.1 we provide an upper bound on the generalization error for approximately margin maximizing algorithms. For convenience we summarize our setting as follows.

**Assumption 2.** *Setting for proving generalization results.*

- *$f : \mathbb{R}^{2m \times d} \times \mathbb{R}^d \to \mathbb{R}$ is a shallow leaky ReLU network as per equation 1.*

- *$\mathcal{A} : \mathbb{R}^{n \times d} \times \{\pm 1\}^n \to \mathbb{R}^{2m \times d}$ is a learning algorithm that returns the weights $\boldsymbol{W} \in \mathbb{R}^{2m \times d}$ of the first layer of $f$.*

- *We let $\boldsymbol{v} \in \mathbb{S}^{d-1}$ and consider training data $(\boldsymbol{X}, \hat{\boldsymbol{y}})$ and test data $(\boldsymbol{x}, y)$ distributed according to $(\boldsymbol{X}, \hat{\boldsymbol{y}}, \boldsymbol{y}, \boldsymbol{x}, y) \sim \mathcal{D}(d, n, k, \gamma, \boldsymbol{v})$ as per Definition 2.1.*

Under this setting we have the following generalization result for an approximately margin maximizing algorithm $\mathcal{A}$. Note this result requires $\gamma$, and hence the signal to noise ratio of the inputs, to be sufficiently large.

**Theorem 3.2.** *Under the setting given in Assumption 2, let $\delta \in (0, 1)$ and suppose $\mathcal{A}$ is approximately margin-maximizing (Definition 2.3). If $n = \Omega\left(\log \frac{1}{\delta}\right)$, $d = \Omega(n)$, $k = O(\frac{n}{1+m|\mathcal{A}|^2})$, and $\gamma = \Omega\left(\frac{1}{k}\right)$ then there is a fixed positive constant $C$ such that with probability at least $1 - \delta$ over $(\boldsymbol{X}, \hat{\boldsymbol{y}})$*

$$\mathbb{P}(yf(\boldsymbol{W}, \boldsymbol{x}) \le 0 \mid \boldsymbol{X}, \hat{\boldsymbol{y}}) \le \exp\left(-C \cdot \frac{d}{k(1+m|\mathcal{A}|^2)}\right).$$

A proof of Theorem 3.2 is provided in Appendix D.2. To comment informally on the relationship between $k$ and $\gamma$, we require $\gamma = \Omega(k^{-1})$ in order to guarantee that any network which achieves zero hinge loss does so by focusing on the signal component $\boldsymbol{v}$ rather than the noise components $\boldsymbol{n}_i$. We use the projection of the model weights onto the signal subspace as a measure of the strength of the signal the model has learned and derive our generalization results based on this measure. In Section 4 we provide a proof sketch of this framework in the simpler, linear model setting. Combining Theorems 3.1, and 3.2 we arrive at the following benign overfitting result for shallow leaky ReLU networks trained with GD on hinge loss.

**Corollary 3.2.1.** *Under the setting given in Assumption 2, let $\delta \in (0, 1)$ and suppose $\mathcal{A} = \mathcal{A}_{GD}$ where $\eta, \lambda \in \mathbb{R}_{>0}$ satisfy Assumption 1. If $n = \Omega\left(\log \frac{1}{\delta}\right)$, $d = \Omega(n)$, $k = O(\alpha^2 n)$, and $\gamma = \Omega\left(\frac{1}{k}\right)$ then the following hold.*

1. *The algorithm $\mathcal{A}_{GD}$ terminates almost surely after a finite number of updates. If $\boldsymbol{W} = \mathcal{A}_{GD}(\boldsymbol{X}, \hat{\boldsymbol{y}})$, then $L(\boldsymbol{W}, \boldsymbol{X}, \hat{\boldsymbol{y}}) = 0$.*

2. *There is a fixed positive constant $C$ such that, with probability at least $1 - \delta$ over the training data $(\boldsymbol{X}, \hat{\boldsymbol{y}})$,*

$$\mathbb{P}(yf(\boldsymbol{W}, \boldsymbol{x}) \leq 0 \mid \boldsymbol{X}, \hat{\boldsymbol{y}}) \leq \exp\left(-C \cdot \frac{\alpha^2 d}{k}\right).$$

We remark that the upper bound is at most $\exp\left(-Cd/n\right)$ for a different constant $C$ as we assume $k = O(\alpha^2 n)$.

If $k$ is large enough, this bound is tight up to constants and factors of $\alpha$ in the exponent. This is given by the following theorem, proven in Appendix D.2.

**Theorem 3.3.** *Under the setting given in Assumption 2, let $\delta \in (0, 1)$ and suppose $\mathcal{A} = \mathcal{A}_{GD}$ where $\eta, \lambda \in \mathbb{R}_{>0}$ satisfy Assumption 1. If $n = \Omega\left(k\right)$, $d = \Omega\left(n\right)$, and $k = \Omega(\log\frac{1}{\delta} + \frac{1}{\alpha})$, then there is a fixed positive constant $C$ such that with probability at least $1 - \delta$ over $(\boldsymbol{X}, \hat{\boldsymbol{y}})$*

$$\mathbb{P}(yf(\boldsymbol{W}, \boldsymbol{x}) \leq 0 \mid \boldsymbol{X}, \hat{\boldsymbol{y}}) \geq \exp\left(-C \cdot \frac{d}{\alpha k}\right).$$

In addition to this benign overfitting result we also provide the following non-benign overfitting result for $\mathcal{A}_{GD}$. Note that conversely this result requires $\gamma$, and hence the signal to noise ratio of the inputs, to be sufficiently small.

**Theorem 3.4.** *Under the setting given in Assumption 2, let $\delta \in (0, 1)$ and suppose $\mathcal{A} = \mathcal{A}_{GD}$, where $\eta, \lambda \in \mathbb{R}_{>0}$ satisfy Assumption 1. If $n = \Omega(1), d = \Omega\left(n + \log\frac{1}{\delta}\right)$ and $\gamma = O\left(\frac{\alpha^3}{d}\right)$ then the following hold.*

1. *The algorithm $\mathcal{A}_{GD}$ terminates almost surely after finitely many updates. With $\boldsymbol{W} = \mathcal{A}_{GD}(\boldsymbol{X}, \hat{\boldsymbol{y}})$, $L(\boldsymbol{W}, \boldsymbol{X}, \hat{\boldsymbol{y}}) = 0$.*

2. *With probability at least $1 - \delta$ over the training data $(\boldsymbol{X}, \hat{\boldsymbol{y}})$*

$$\mathbb{P}(yf(\boldsymbol{W}, \boldsymbol{x}) < 0 \mid \boldsymbol{X}, \hat{\boldsymbol{y}}) \geq \frac{1}{8}.$$

A proof of Theorem 3.4 is provided in Appendix D.3.

## 4    Approximate margin maximization and generalization: insight from linear models

In this section we outline proofs for the analogues of Theorems 3.2 and 3.4 in the context of linear models. The arguments are thematically similar and clearer to present. We provide complete proofs of benign and non-benign overfitting for linear models in Appendix C.

An important lemma is the following, which bounds the largest and $n$-th largest singular values ($\sigma_1$ and $\sigma_n$ respectively) of the noise matrix $\boldsymbol{N}$:

**Lemma 4.1.** *Let $\boldsymbol{N} \in \mathbb{R}^{n \times d}$ denote a random matrix whose rows are drawn mutually i.i.d. from $\mathcal{N}(\boldsymbol{0}_d, \frac{1}{d}(\boldsymbol{I}_d - \boldsymbol{v}\boldsymbol{v}^T))$. If $d = \Omega\left(n + \log\frac{1}{\delta}\right)$, then there exists constants $C_1$ and $C_2$ such that, with probability at least $1 - \delta$,*

$$C_1 \leq \sigma_n(\boldsymbol{N}) \leq \sigma_1(\boldsymbol{N}) \leq C_2.$$

We prove this lemma in Appendix B using results from Vershynin (2018) and Rudelson & Vershynin (2009). A consequence of this lemma is that with probability at least $1 - \delta$, the condition number of $\boldsymbol{N}$ restricted to $\mathrm{span}\,\boldsymbol{N}$ can be bounded above independently of all hyperparameters. For this reason, we refer to the noise as being well-conditioned.

Now let $\boldsymbol{w} = \mathcal{A}(\boldsymbol{X}, \hat{\boldsymbol{y}})$ be the linear classifier returned by the algorithm. Observe that we can decompose the weight vector into a signal and noise component

$$\boldsymbol{w} = a_v\boldsymbol{v} + \boldsymbol{z},$$

where $\boldsymbol{z} \perp \boldsymbol{v}$ and $a_v \in \mathbb{R}$. Based on this decomposition the proof proceeds as follows.

**1. Generalization bounds based on the SNR:** For test data as per the data model given in Definition 2.1 we want to bound the probability of misclassification: in particular, we want to bound the probability that

$$X := y\langle \boldsymbol{w}, \boldsymbol{x}\rangle = \sqrt{\gamma}a_v + \sqrt{1-\gamma}\langle \boldsymbol{n}, \boldsymbol{z}\rangle \leq 0.$$

As the noise is normally distributed, $X \sim \mathcal{N}\left(\sqrt{\gamma}a_v, \frac{1-\gamma}{d}\|\boldsymbol{z}\|^2\right)$ and the desired upper bound therefore follows from Hoeffding's inequality,

$$\mathbb{P}(X \leq 0) \leq \exp\left(-\frac{\gamma d a_v^2}{2(1-\gamma)\|\boldsymbol{z}\|^2}\right).$$

Using Gaussian anti-concentration, we also obtain a lower bound for the probability of misclassification:

$$\mathbb{P}(y\langle \boldsymbol{w}, \boldsymbol{x}\rangle \leq 0) \geq \max\left\{\frac{1}{2} - \sqrt{\frac{d\gamma}{2\pi(1-\gamma)}}\frac{a_v}{\|\boldsymbol{z}\|}, \frac{1}{4}\exp\left(-\frac{6d}{\pi}\frac{\gamma}{1-\gamma}\frac{a_v^2}{\|\boldsymbol{z}\|^2}\right)\right\}.$$

**2. Upper bound the norm of the max-margin classifier:** In order to use the approximate max-margin property we require an upper bound on $\|\boldsymbol{w}^*\|$. As by definition $\|\boldsymbol{w}^*\| \leq \|\tilde{\boldsymbol{w}}\|$, it suffices to construct a vector $\tilde{\boldsymbol{w}}$ that interpolates the data and has small norm. Using that the noise matrix of the data is well-conditioned with high probability, we achieve this by strategically constructing the signal and noise components of $\tilde{\boldsymbol{w}}$. This yields the bound

$$\|\boldsymbol{w}^*\| \leq \|\tilde{\boldsymbol{w}}\| \leq C\min\left(\sqrt{\frac{n}{1-\gamma}}, \sqrt{\frac{1}{\gamma} + \frac{k}{1-\gamma}}\right),$$

where the arguments of the min function originate from a small and large $\gamma$ regime respectively.

**3. Lower bound the SNR using the approximate margin maximization property:** Based on step 1 the key quantity of interest from a generalization perspective is the ratio $a_v/\|\boldsymbol{z}\|$, which describes the signal to noise ratio (SNR) of the learned classifier. To lower bound this quantity we first lower bound $a_v$. In particular, if $a_v$ is small, then the only way to attain zero loss on the clean data is for $\|\boldsymbol{z}\|$ to be large. However, under appropriate assumptions on $d, n, k$ and $\gamma$ this can be shown to contradict the bound $\|\boldsymbol{z}\| \leq \|\boldsymbol{w}\| \leq |\mathcal{A}|\|\boldsymbol{w}^*\|$, and thus $a_v$ must be bounded from below. A lower bound on $a_v/\|\boldsymbol{z}\|$ then follows by again using $\|\boldsymbol{z}\| \leq \|\boldsymbol{w}^*\|$. Hence we obtain a lower bound for the SNR and establish benign overfitting.

**4. Upper bound the SNR using the zero loss condition:** For the generalization lower bound, we compute an upper bound for the ratio $a_v/\|\boldsymbol{z}\|$ rather than a lower bound. Since the model perfectly fits the training data with margin one,

$$1 \leq \hat{y}_i\langle \boldsymbol{w}, \boldsymbol{x}_i\rangle = \sqrt{\gamma}\beta_i a_v + \sqrt{1-\gamma}\hat{y}_i\langle \boldsymbol{z}, \boldsymbol{n}_i\rangle$$

for all $i \in [n]$. The above inequality implies that $\sqrt{1-\gamma}\hat{y}_i\langle \boldsymbol{n}_i, \boldsymbol{z}\rangle$ is at least $\sqrt{\gamma}a_v$ for all corrupt points. Since the noise is well-conditioned, this gives a lower bound on $\|\boldsymbol{z}\|$ in terms of $a_v$ and hence an upper bound on the SNR $a_v/\|\boldsymbol{z}\|$. By the second generalization lower bound in step 1, the generalization error is bounded below at a similar exponential rate to the upper bound.

**5. Upper bound the SNR using the zero loss condition and maximum margin property:** To prove non-benign overfitting, we again compute an upper bound for the ratio $a_v/\|\boldsymbol{z}\|$. We return to the zero loss condition:

$$1 \leq \hat{y}_i\langle \boldsymbol{w}, \boldsymbol{x}_i\rangle = \sqrt{\gamma}\beta_i a_v + \sqrt{1-\gamma}\hat{y}_i\langle \boldsymbol{z}, \boldsymbol{n}_i\rangle$$

for all $i \in [n]$. If $\gamma$ is small and $|\sqrt{\gamma}a_v|$ is large, then $\|\boldsymbol{w}\|$ will be large, contradicting the approximate margin maximization. Hence the above inequality implies that $\sqrt{1-\gamma}\hat{y}_i\langle \boldsymbol{n}_i, \boldsymbol{z}\rangle$ is large for all $i \in [n]$. Since the noise is well-conditioned, this can only happen when $\|\boldsymbol{z}\|$ is large. This gives us a lower bound on $\|\boldsymbol{z}\|$. As before, we can also upper bound $a_v$ by $\|\boldsymbol{w}\|$, giving us an upper bound on the SNR $a_v/\|\boldsymbol{z}\|$. By the first generalization lower bound in step 1, the classifier generalizes poorly and exhibits non-benign overfitting.

## 4.1 From linear models to leaky ReLU networks

The proof of benign and non-benign overfitting in the linear case uses the tension between the two properties of approximate margin maximization: fitting both the clean and corrupt points with margin versus the bound on the norm. To extend this idea to a shallow leaky ReLU network as per equation 1, we consider the same decomposition for each neuron $j \in [2m]$,

$$\boldsymbol{w}_j = a_j \boldsymbol{v} + \boldsymbol{z}_j,$$

where $a_j \in \mathbb{R}$ and $\boldsymbol{z}_j \perp \boldsymbol{v}$. In the linear case $\pm a_v$ can be interpreted as the activation of the linear classifier on $\pm \boldsymbol{v}$ respectively: in terms of magnitude the signal activation is the same in either case and thus we measure the alignment of the linear model with the signal using $|a_v|$. For leaky ReLU networks we define their activation on $\pm \boldsymbol{v}$ respectively as $A_1 = f(\boldsymbol{W}, \boldsymbol{v})$ and $A_{-1} = -f(\boldsymbol{W}, -\boldsymbol{v})$, and then define the alignment of the network as $A_{\min} = \min\{A_1, A_{-1}\}$. Considering the alignment of the network with the noise, then if $\boldsymbol{Z} \in \mathbb{R}^{2m \times d}$ denotes a matrix whose $j$-th row is $\boldsymbol{z}_j$, then we measure the alignment of the network using $\|\boldsymbol{Z}\|_F$. As a result, analogous to $a_v/\|\boldsymbol{z}\|$, the key ratio from a generalization perspective in the context of a leaky ReLU network is $A_{\min}/\|\boldsymbol{Z}\|_F$. The proof Theorems 3.2, 3.3, and 3.4 then follow the same outline as Steps 1-3 above but with additional non-trivial technicalities.

## 5 Conclusion

In this work we have proven conditions under which leaky ReLU networks trained on binary classification tasks exhibit benign and non-benign overfitting. We have substantially relaxed the necessary assumptions on the input data compared with prior work; instead of requiring nearly orthogonal data with $d = \Omega(n^2 \log n)$ or higher, we only need $d = \Omega(n)$. We achieve this by using the distribution of singular values of the noise rather than specific correlations between noise vectors. Our emphasis was on networks trained by gradient descent with the hinge loss, but we establish a new framework that is general enough to accommodate any algorithm that is approximately margin maximizing.

There are a few limitations of our results which would be natural questions to address in future work. While we improve upon existing results in our dependence on the input dimension of the data, we still require that the training dataset is linearly separable. This leaves open the question of whether an overparameterized network will perfectly fit the training data and generalize well for lower dimensional data, or satisfy a similar margin maximization condition. We also focus mainly on two-layer networks with fixed outer layer weights trained with the hinge loss. It would be interesting to investigate whether analogous results hold for deeper architectures or different loss functions and data models.

### Acknowledgments

This material is based upon work supported by the National Science Foundation under Grant No. DMS-1928930 and by the Alfred P. Sloan Foundation under grant G-2021-16778, while the authors EG and DN were in residence at the Simons Laufer Mathematical Sciences Institute (formerly MSRI) in Berkeley, California, during the Fall 2023 semester. EG and DN were also partially supported by NSF DMS 2011140. EG was also supported by a NSF Graduate Research Fellowship under grant DGE 2034835. GM and KK were partly supported by NSF CAREER DMS 2145630 and DFG SPP 2298 Theoretical Foundations of Deep Learning grant 464109215. GM was also partly supported by NSF grant CCF 2212520, ERC Starting Grant 757983 (DLT), and BMBF in DAAD project 57616814 (SECAI).

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

# Appendix A  Preliminaries on random vectors

Recall that the sub-exponential norm of a random variable $X$ is defined as
$$\|X\|_{\psi_1} := \inf\{t > 0 \colon \mathbb{E}[\exp(|X|/t)] \le 2\}$$
(see Vershynin, 2018, Definition 2.7.5) and that the sub-Gaussian norm is defined as
$$\|X\|_{\psi_2} := \inf\{t > 0 \colon \mathbb{E}[\exp(X^2/t^2)] \le 2\}.$$
A random variable $X$ is sub-Gaussian if and only if $X^2$ is sub-exponential. Furthermore, $\|X^2\|_{\psi_1} = \|X\|_{\psi_2}^2$.

**Lemma A.1.** *Let $\boldsymbol{n} \sim \mathcal{N}(\boldsymbol{0}_d, d^{-1}(\boldsymbol{I}_d - \boldsymbol{v}\boldsymbol{v}^T))$ and suppose that $\boldsymbol{Z} \in \mathbb{R}^{m \times d}$. Then with probability at least $1 - \epsilon$,*
$$\|\boldsymbol{Z}\boldsymbol{n}\| \le C\|\boldsymbol{Z}\|_F \sqrt{\frac{1}{d}\log\frac{1}{\epsilon}}.$$

*Proof.* Let $\boldsymbol{P} = \boldsymbol{I}_d - \boldsymbol{v}\boldsymbol{v}^T$ be the orthogonal projection onto $\mathrm{span}(\{\boldsymbol{v}\})^\perp$, so that $\boldsymbol{Z}\boldsymbol{n}$ is identically distributed to $d^{-1/2}\boldsymbol{Z}\boldsymbol{P}\boldsymbol{n}'$, where $\boldsymbol{n}'$ has distribution $\mathcal{N}(\boldsymbol{0}_d, \boldsymbol{I}_d)$. Following Vershynin (2018, Theorem 6.3.2),
$$\begin{aligned}
\left\| \|\boldsymbol{Z}\boldsymbol{n}\| - \|d^{-1/2}\boldsymbol{Z}\boldsymbol{P}\|_F \right\|_{\psi_2} &= \left\| \|d^{-1/2}\boldsymbol{Z}\boldsymbol{P}\boldsymbol{n}'\| - \|d^{-1/2}\boldsymbol{Z}\boldsymbol{P}\|_F \right\|_{\psi_2} \\
&\le C\|d^{-1/2}\boldsymbol{Z}\boldsymbol{P}\| \\
&\le Cd^{-1/2}\|\boldsymbol{Z}\|\|\boldsymbol{P}\| \\
&= Cd^{-1/2}\|\boldsymbol{Z}\| \\
&\le Cd^{-1/2}\|\boldsymbol{Z}\|_F,
\end{aligned}$$
where we used that $\boldsymbol{P}$ is an orthogonal projection in the fourth line and that the operator norm is by bounded above by the Frobenius norm in the fifth line. As a result the sub-Gaussian norm of $\|\boldsymbol{Z}\boldsymbol{n}\|$ is bounded as
$$\begin{aligned}
\left\| \|\boldsymbol{Z}\boldsymbol{n}\| \right\|_{\psi_2} &\le \left\| \|\boldsymbol{Z}\boldsymbol{n}\| - \|d^{-1/2}\boldsymbol{Z}\boldsymbol{P}\|_F \right\|_{\psi_2} + \left\| \|d^{-1/2}\boldsymbol{Z}\boldsymbol{P}\|_F \right\|_{\psi_2} \\
&\le Cd^{-1/2}(\|\boldsymbol{Z}\|_F + \|\boldsymbol{Z}\boldsymbol{P}\|_F) \\
&\le Cd^{-1/2}\|\boldsymbol{Z}\|_F,
\end{aligned}$$
where the last line follows from the calculation
$$\begin{aligned}
\|\boldsymbol{Z}\boldsymbol{P}\|_F &= \|\boldsymbol{P}^T\boldsymbol{Z}^T\|_F \\
&= \|\boldsymbol{P}\boldsymbol{Z}^T\|_F \\
&\le \|\boldsymbol{P}\|\|\boldsymbol{Z}^T\|_F \\
&= \|\boldsymbol{Z}^T\|_F \\
&= \|\boldsymbol{Z}\|_F.
\end{aligned}$$
This implies a tail bound (see Vershynin, 2018, Proposition 2.5.2)
$$\mathbb{P}(\|\boldsymbol{Z}\boldsymbol{n}\| \ge t) \le 2\exp\left(-\frac{dt^2}{C\|\boldsymbol{Z}\|_F^2}\right), \quad \text{for all } t \ge 0.$$

Setting $t = C\|\boldsymbol{Z}\|_F\sqrt{\frac{1}{d}\log\frac{2}{\epsilon}}$, the result follows. $\qquad\square$

**Lemma A.2.** *Let $\boldsymbol{n} \sim \mathcal{N}(\boldsymbol{0}_d, d^{-1}(\boldsymbol{I}_d - \boldsymbol{v}\boldsymbol{v}^T))$ and suppose $\boldsymbol{z} \in \mathrm{span}(\{\boldsymbol{v}\})^\perp$. There exists a $C > 0$ such that with probability at least $1 - \delta$*
$$|\langle \boldsymbol{n}, \boldsymbol{z}\rangle| \le C\|\boldsymbol{z}\|\sqrt{\frac{1}{d}\log\frac{1}{\delta}}.$$

*Furthermore, there exists a $c > 0$ such that with probability at least $\frac{1}{2}$*
$$|\langle \boldsymbol{n}, \boldsymbol{z}\rangle| \ge \frac{c\|\boldsymbol{z}\|}{\sqrt{d}}.$$

*Proof.* Let $X = \langle \boldsymbol{n}, \boldsymbol{z} \rangle$. Then $X$ is Gaussian with variance

$$
\begin{aligned}
\mathbb{E}[X^2] &= \mathbb{E}[\boldsymbol{z}^T \boldsymbol{n} \boldsymbol{n}^T \boldsymbol{z}] \\
&= \boldsymbol{z}^T \mathbb{E}[\boldsymbol{n} \boldsymbol{n}^T] \boldsymbol{z} \\
&= d^{-1} \boldsymbol{z}^T (\boldsymbol{I}_d - \boldsymbol{v} \boldsymbol{v}^T) \boldsymbol{z} \\
&= d^{-1} \boldsymbol{z}^T \boldsymbol{z} \\
&= d^{-1} \|\boldsymbol{z}\|^2.
\end{aligned}
$$

Note the third line above follows from the fact that $\boldsymbol{z} \in \mathrm{span}(\{\boldsymbol{v}\})^\perp$. Therefore by Hoeffding's inequality, for all $t \geq 0$

$$
\mathbb{P}(|\langle \boldsymbol{n}, \boldsymbol{z} \rangle| \geq t) \leq 2 \exp\left( -\frac{dt^2}{C\|\boldsymbol{z}\|^2} \right).
$$

Setting $t = C\|\boldsymbol{z}\|^2 \sqrt{\frac{1}{d} \log \frac{2}{\delta}}$, we obtain

$$
\mathbb{P}(|\langle \boldsymbol{n}, \boldsymbol{z} \rangle| \geq t) \leq \delta,
$$

which establishes the first part of the result.

Since $d^{1/2} \|\boldsymbol{z}\|^{-1} X$ is a standard Gaussian, there exists a constant $c$ such that

$$
\mathbb{P}(|d^{1/2} \|\boldsymbol{z}\|^{-1} X| \geq c) \leq \frac{1}{2}.
$$

Rearranging, we obtain

$$
\mathbb{P}(|\langle \boldsymbol{n}, \boldsymbol{z} \rangle| \geq cd^{-1/2} \|\boldsymbol{z}\|) \leq \frac{1}{2}.
$$

This establishes the second part of the result. $\qquad \square$

## Appendix B    Upper bounding the norm of the max-margin classifier of the data

Here we establish key properties concerning the data model given in Definition 2.1, our main goal being to establish bounds on the norm of the max-margin classifier. To this end we first identify certain useful facts about rectangular Gaussian matrices. In what follows we index the singular values of any given matrix $\boldsymbol{A}$ in decreasing order as $\sigma_1(A) \geq \sigma_2(A) \geq \cdots$. Furthermore, we denote the $i$-th row of a matrix $\boldsymbol{A}$ as $\boldsymbol{a}_i$.

**Lemma B.1.** *Let $\boldsymbol{G} \in \mathbb{R}^{n \times d}$ be a Gaussian matrix whose entries are mutually i.i.d. with distribution $\mathcal{N}(0, 1)$. If $d = \Omega\left(n + \log \frac{1}{\delta}\right)$, then with probability at least $1 - \delta$ the following inequalities are simultaneously true.*

1. $\sigma_1(\boldsymbol{G}) \leq C(\sqrt{d} + \sqrt{n})$,

2. $\sigma_n(\boldsymbol{G}) \geq c(\sqrt{d} - \sqrt{n})$.

*Proof.* We proceed by upper bounding the probability that each individual inequality does not hold.

*1.* To derive an upper bound on $\sigma_1(\boldsymbol{G})$ we use the following fact (see Vershynin, 2018, Theorem 4.4.5). For any $\epsilon > 0$,

$$
\mathbb{P}(\sigma_1(\boldsymbol{G}) \geq C_1(\sqrt{n} + \sqrt{d} + \epsilon)) \leq 2 \exp(-\epsilon^2).
$$

With $\epsilon = \sqrt{n} + \sqrt{d}$ and $d \geq \log \frac{4}{\delta}$ then

$$
\begin{aligned}
\mathbb{P}(\sigma_1(\boldsymbol{G}) \geq 2C_1(\sqrt{n} + \sqrt{d})) &\leq 2 \exp(-d) \\
&\leq \frac{\delta}{2}.
\end{aligned}
$$

2. To derive a lower bound on $\sigma_n(\boldsymbol{G})$ we use the following fact (see Rudelson & Vershynin, 2009, Theorem 1.1). There exist constants $C_1, C_2 > 0$ such that, for any $\epsilon > 0$,

$$\mathbb{P}(\sigma_n(\boldsymbol{G}) \leq \epsilon(\sqrt{d} - \sqrt{n-1})) \leq (C_1\epsilon)^{d-n+1} + e^{-C_2 d}.$$

Let $\epsilon = \frac{1}{C_1 e}$ and let $d \geq 2n + \left(2 + \frac{1}{C_2}\right)\log\frac{4}{\delta}$. Then

$$\mathbb{P}(\sigma_n(\boldsymbol{A}) \leq \epsilon(\sqrt{d} - \sqrt{n})) \leq \exp(-d/2) + \exp(-C_2 d)$$
$$\leq \frac{\delta}{4} + \frac{\delta}{4}$$
$$= \frac{\delta}{2}.$$

Hence both bounds hold simultaneously with probability at least $1 - \delta$. $\qquad\square$

The next lemma formulates lower and upper bounds on the smallest and largest singular values of a noise matrix under our data model.

**Lemma 4.1.** *Let $\boldsymbol{N} \in \mathbb{R}^{n \times d}$ denote a random matrix whose rows are drawn mutually i.i.d. from $\mathcal{N}(\boldsymbol{0}_d, \frac{1}{d}(\boldsymbol{I}_d - \boldsymbol{v}\boldsymbol{v}^T))$. If $d = \Omega\left(n + \log\frac{1}{\delta}\right)$, then there exists constants $C_1$ and $C_2$ such that, with probability at least $1 - \delta$,*

$$C_1 \leq \sigma_n(\boldsymbol{N}) \leq \sigma_1(\boldsymbol{N}) \leq C_2.$$

*Proof.* Let $\mathbb{H} = \text{span}(\{\boldsymbol{v}\})^\perp \cong \mathbb{R}^{d-1}$. Let $\boldsymbol{N}' : \mathbb{H} \to \mathbb{R}^n$ be a random matrix whose rows are drawn mutually i.i.d. from $\mathcal{N}(\boldsymbol{0}_d, \boldsymbol{I}_\mathbb{H})$. Since $d = \Omega\left(n + \log\frac{1}{\delta}\right)$, with probability at least $1 - \delta$, we have

$$c(\sqrt{d} - \sqrt{n}) \leq \sigma_n(\boldsymbol{N}') \leq \sigma_1(\boldsymbol{N}') \leq C(\sqrt{d} + \sqrt{n}).$$

We denote the above event by $\omega$. Let $\boldsymbol{J} : \mathbb{H} \to \mathbb{R}^d$ be the inclusion map and let $\boldsymbol{P} = \boldsymbol{I}_d - \boldsymbol{v}\boldsymbol{v}^T$. For any random vector $\boldsymbol{n}$ with distribution $\mathcal{N}(\boldsymbol{0}_d, \boldsymbol{I}_\mathbb{H})$, $\boldsymbol{J}\boldsymbol{n}$ is a Gaussian random vector with covariance matrix $\boldsymbol{J}\boldsymbol{J}^T = \boldsymbol{P}$. Therefore, $d^{-1/2}\boldsymbol{N}'\boldsymbol{J}^T$ is a random matrix whose rows are drawn mutually i.i.d. from $\mathcal{N}(\boldsymbol{0}_d, d^{-1}\boldsymbol{P})$. That is, $d^{-1/2}\boldsymbol{N}'\boldsymbol{J}^T$ is identically distributed to $\boldsymbol{N}$. For the lower bound on the $n$-th largest singular value, if $d \geq \frac{4n}{c^2}$, then conditional on $\omega$ we have

$$\sigma_n(d^{-1/2}\boldsymbol{N}'\boldsymbol{J}^T) = d^{-1/2}\sigma_{\min}(\boldsymbol{J}\boldsymbol{N}'^T)$$
$$\geq d^{-1/2}\sigma_{\min}(\boldsymbol{J})\sigma_{\min}(\boldsymbol{N}'^T)$$
$$= d^{-1/2}\sigma_{\min}(\boldsymbol{N}'^T)$$
$$= d^{-1/2}\sigma_n(\boldsymbol{N}')$$
$$\geq c - \sqrt{\frac{n}{d}}$$
$$\geq \frac{c}{2}.$$

Note here we define $\sigma_{\min}$ to be the smallest singular value of a matrix. In the first line we used $\boldsymbol{J}\boldsymbol{N}'^T$ is a linear map $\mathbb{R}^n \to \mathbb{R}^d$ and $d \geq n$, and in the third line we used the fact that $\boldsymbol{J}$ is an inclusion map. For the upper bound on the largest singular value, if $d \geq \frac{n}{C^2}$, then conditional on $\omega$ we have

$$\sigma_1(d^{-1/2}\boldsymbol{N}'\boldsymbol{J}^T) = d^{-1/2}\sigma_1(\boldsymbol{J}\boldsymbol{N}'^T)$$
$$\leq d^{-1/2}\sigma_1(\boldsymbol{N}'^T)$$
$$= d^{-1/2}\sigma_1(\boldsymbol{N}')$$
$$\leq C + \sqrt{\frac{n}{d}}$$
$$\leq 2C.$$

Note here that again we used the fact that $\boldsymbol{J}$ is an inclusion map in the first line. Therefore, if $d = \Omega\left(n + \log\frac{1}{\delta}\right)$

$$\mathbb{P}\left(\frac{c}{2} \leq \sigma_n(\boldsymbol{N}) \leq \sigma_1(\boldsymbol{N}) \leq 2C\right) \geq \mathbb{P}(\omega)$$
$$\geq 1 - \delta.$$

$\square$

The following lemma is useful for constructing vectors in the noise subspace with properties suitable for bounding the norm of the max-margin solution. We remark that the same approach could be used in the setting where the noise and signal are not orthogonal by considering the pseudo-inverse $([\boldsymbol{N}, \boldsymbol{v}]^T)^\dagger$ instead of $\boldsymbol{N}^\dagger$.

**Lemma B.2.** *Let $\mathcal{I} \subseteq [n]$ be an arbitrary subset such that $|\mathcal{I}| = \ell$. In the context of the data model given in Definition 2.1, assume $d = \Omega\left(n + \log\frac{1}{\delta}\right)$. Then there exists $\boldsymbol{z} \in \mathbb{R}^d$ such that with probability at least $1 - \delta$ the following hold simultaneously.*

1. *$\hat{y}_i\langle \boldsymbol{n}_i, \boldsymbol{z}\rangle = 1$ for all $i \in \mathcal{I}$,*

2. *$\hat{y}_i\langle \boldsymbol{n}_i, \boldsymbol{z}\rangle = 0$ for all $i \notin \mathcal{I}$,*

3. *$\boldsymbol{z} \perp \boldsymbol{v}$,*

4. *$C_1 \leq \|\boldsymbol{z}\| \leq C_2$.*

*Proof.* Recall that $\boldsymbol{N} \in \mathbb{R}^{n \times d}$ is a random matrix whose rows are selected i.i.d. from $\mathcal{N}(\boldsymbol{0}_d, d^{-1}(\boldsymbol{I}_d - \boldsymbol{v}\boldsymbol{v}^T))$. By Lemma 4.1, with probability at least $1 - \delta$ we have

$$c \leq \sigma_n(\boldsymbol{N}) \leq \sigma_1(\boldsymbol{N}) \leq C.$$

Conditioning on this event, we will construct a vector $\boldsymbol{z}$ which satisfies the desired properties. Let $\boldsymbol{w} \in \mathbb{R}^n$ satisfy $w_i = \hat{y}_i$ if $i \in \mathcal{I}$ and $w_i = 0$ otherwise. Let $\boldsymbol{z} = \boldsymbol{N}^\dagger \boldsymbol{w}$, where $\boldsymbol{N}^\dagger = \boldsymbol{N}^T(\boldsymbol{N}\boldsymbol{N}^T)^{-1}$ is the right pseudo-inverse of $\boldsymbol{N}$. Then $\boldsymbol{N}\boldsymbol{z} = \boldsymbol{w}$. In particular, for $i \in \mathcal{I}$, $\hat{y}_i\langle \boldsymbol{n}_i, \boldsymbol{z}\rangle = \hat{y}_i w_i = 1$, and for $i \notin \mathcal{I}$, $\hat{y}_i\langle \boldsymbol{n}_i, \boldsymbol{z}\rangle = \hat{y}_i w_i = 0$. This establishes properties 1 and 2. Since $\boldsymbol{N}^\dagger \boldsymbol{w}$ is in the span of the set $\{\boldsymbol{n}_i\}_{i \in [n]}$, it is orthogonal to $\boldsymbol{v}$. This establishes property 3. Finally, we can bound

$$\begin{aligned}
\|\boldsymbol{z}\| &= \|\boldsymbol{N}^\dagger \boldsymbol{w}\| \\
&\leq \|\boldsymbol{N}^\dagger\|\|\boldsymbol{w}\| \\
&= \frac{\|\boldsymbol{w}\|}{\sigma_n(\boldsymbol{N})} \\
&\leq \frac{\|\boldsymbol{w}\|}{c} \\
&= \frac{\sqrt{\ell}}{c}
\end{aligned}$$

and

$$\begin{aligned}
\|\boldsymbol{z}\| &= \|\boldsymbol{N}^\dagger \boldsymbol{w}\| \\
&\geq \sigma_n(\boldsymbol{N}^\dagger)\|\boldsymbol{w}\| \\
&= \frac{\|\boldsymbol{w}\|}{\sigma_1(\boldsymbol{N})} \\
&\geq \frac{\|\boldsymbol{w}\|}{C} \\
&= \frac{\sqrt{\ell}}{C}
\end{aligned}$$

which establishes property 4.

$\square$

With Lemma B.2 in place we are now able to appropriately bound the max-margin norm.

**Lemma B.3.** *In the context of the data model given in Definition 2.1, let $\boldsymbol{w}^*$ denote the max-margin classifier of the training data $(\boldsymbol{X}, \hat{\boldsymbol{y}})$, which exists almost surely. If $d = \Omega\left(n + \log\frac{1}{\delta}\right)$ then with probability at least $1 - \delta$*

$$\|\boldsymbol{w}^*\| \le C\sqrt{\frac{1}{\gamma} + \frac{k}{1 - \gamma}},$$

*where $C > 0$ is a constant.*

*Proof.* Under the assumptions stated, the conditions of Lemma B.2 hold with probability at least $1 - \delta$. Conditioning on this let

$$\boldsymbol{w} = \frac{1}{\sqrt{\gamma}}\boldsymbol{v} + \frac{2}{\sqrt{1 - \gamma}}\boldsymbol{z}$$

where $\boldsymbol{z}$ is the vector constructed in Lemma B.2 with $\mathcal{I} = \mathcal{B}$. For any $i \in [n]$ we therefore have

$$\hat{y}_i \langle \boldsymbol{x}_i, \boldsymbol{w} \rangle = \beta_i + 2\hat{y}_i \langle \boldsymbol{n}_i, \boldsymbol{z} \rangle.$$

As a result, for $i \in \mathcal{G}$

$$\hat{y}_i \langle \boldsymbol{x}_i, \boldsymbol{w} \rangle = 1 + 2\hat{y}_i \langle \boldsymbol{n}_i, \boldsymbol{z} \rangle = 1,$$

while for $l \in \mathcal{B}$

$$\hat{y}_i \langle \boldsymbol{x}_i, \boldsymbol{w} \rangle = -1 + 2\hat{y}_i \langle \boldsymbol{n}_i, \boldsymbol{z} \rangle = 1.$$

As a result $\hat{y}_i \langle \boldsymbol{x}_i, \boldsymbol{w} \rangle = 1$ for all $i \in [n]$. Furthermore, observe

$$\|\boldsymbol{w}\|^2 = \frac{1}{\gamma} + \frac{4}{1 - \gamma}\|\boldsymbol{z}\|^2$$

$$\le C\left(\frac{1}{\gamma} + \frac{k}{1 - \gamma}\right)$$

for a universal constant $C$. To conclude observe $\|\boldsymbol{w}^*\| \le \|\boldsymbol{w}\|$ by definition of being max-margin. $\qquad\square$

Lemma B.3 constructs a classifier with margin one using an appropriate linear combination of the signal vector $\boldsymbol{v}$ and a vector in the noise subspace which classifies all noise components belonging to bad points correctly. This bound is useful for the benign overfitting setting in which $\gamma$ is not too small. However, for small $\gamma$, as is the case in the non-benign overfitting setting, this bound behaves poorly as the only way the construction can fit all the data points is by making the coefficient in front of the $\boldsymbol{v}$ component large. For the non-benign overfitting setting we therefore require a different approach and instead fit all data points based on their noise components alone. In particular, the following bound behaves better than that given in Lemma B.3 when $\gamma$ approaches 0.

**Lemma B.4.** *In the context of the data model given in Definition 2.1, let $\boldsymbol{w}^*$ denote the max-margin classifier of the training data $(\boldsymbol{X}, \hat{\boldsymbol{y}})$, which exists almost surely. If $d = \Omega\left(n + \log\frac{1}{\delta}\right)$ then with probability at least $1 - \delta$*

$$\|\boldsymbol{w}^*\| \le C\sqrt{\frac{n}{1 - \gamma}}.$$

*Proof.* Applying Lemma B.2 with $\mathcal{I} = [n]$ then with probability $1 - \delta$ there exists $\boldsymbol{z} \in \mathbb{R}^d$ such that $\|\boldsymbol{z}\| = \Theta(\sqrt{n})$, $\hat{y}_i \langle \boldsymbol{n}_i, \boldsymbol{z} \rangle = 1$ for all $i \in [n]$ and $z \perp v$. Conditioning on this event, let

$$\boldsymbol{w} = \frac{1}{\sqrt{1 - \gamma}}\boldsymbol{z}.$$

Then for all $i \in [n]$,

$$\hat{y}_i \langle \boldsymbol{x}_i, \boldsymbol{w} \rangle = \hat{y}_i \langle \boldsymbol{n}_i, \boldsymbol{z} \rangle = 1.$$

Furthermore, there exists a constant $C > 0$ such that

$$\|\boldsymbol{w}\| \le C\sqrt{\frac{n}{1 - \gamma}}.$$

To conclude observe $\|\boldsymbol{w}^*\| \le \|\boldsymbol{w}\|$ by definition of being max-margin. $\qquad\square$

# Appendix C  Linear models

## C.1  Sufficient conditions for benign and harmful overfitting

We start by providing a lemma which characterizes the generalization properties of linear classifiers.

**Lemma C.1.** *In the context of the data model given in Definition 2.1, consider the linear classifier* $w = a_v v + z$, *where* $a_v \in \mathbb{R}$ *and* $\langle z, v \rangle = 0$. *If* $a_v \geq 0$, *then the generalization error can be bounded as follows:*

$$\mathbb{P}(y\langle w, x \rangle \leq 0) \leq \exp\left(-\frac{d}{2}\frac{\gamma}{1-\gamma}\frac{a_v^2}{\|z\|^2}\right).$$

*and*

$$\mathbb{P}(y\langle w, x \rangle \leq 0) \geq \max\left\{\frac{1}{2} - \sqrt{\frac{d\gamma}{2\pi(1-\gamma)}}\frac{a_v}{\|z\|}, \frac{1}{4}\exp\left(-\frac{6d}{\pi}\frac{\gamma}{1-\gamma}\frac{a_v^2}{\|z\|^2}\right)\right\}.$$

*Proof.* Recall from Definition 2.1 that a test pair $(x, y)$ satisfies

$$x = y(\sqrt{\gamma}v + \sqrt{1-\gamma}n),$$

where $n \sim \mathcal{N}(0_d, d^{-1}(I_d - vv^T))$ is a random vector. Let $X = y\langle w, x \rangle$, so

$$X = \langle a_v v + z, \sqrt{\gamma}v + \sqrt{1-\gamma}n \rangle$$
$$= \sqrt{\gamma}a_v + \sqrt{1-\gamma}\langle n, z \rangle.$$

Then $X$ is a Gaussian random variable with expectation

$$\mathbb{E}[X] = \sqrt{\gamma}a_v + \sqrt{1-\gamma}\mathbb{E}[\langle n, z \rangle]$$
$$= \sqrt{\gamma}a_v$$

and variance

$$\text{Var}(X) = (1-\gamma)\text{Var}(\langle n, z \rangle)$$
$$= (1-\gamma)\mathbb{E}[z^T n n^T z]$$
$$= \frac{1-\gamma}{d}z^T(I_d - vv^T)z$$
$$= \frac{1-\gamma}{d}z^T z$$
$$= \frac{(1-\gamma)\|z\|^2}{d}.$$

By Hoeffding's inequality, for all $t \geq 0$,

$$\mathbb{P}(X \leq \sqrt{\gamma}a_v - t) \leq \exp\left(-\frac{t^2 d}{2(1-\gamma)\|z\|^2}\right).$$

Setting $t = \sqrt{\gamma}a_v$, we obtain

$$\mathbb{P}(X \leq 0) \leq \exp\left(-\frac{\gamma d a_v^2}{2(1-\gamma)\|z\|^2}\right),$$

which establishes the upper bound on the generalization error.

To prove the lower bound, we integrate a standard Gaussian pdf:

$$\mathbb{P}(X \leq 0) = \mathbb{P}\left(\frac{X - \sqrt{\gamma}a_v}{\sqrt{1-\gamma}\|z\|/\sqrt{d}} \leq -\sqrt{\frac{\gamma d}{1-\gamma}}\frac{a_v}{\|z\|}\right)$$
$$= \frac{1}{2} - \frac{1}{\sqrt{2\pi}}\int_{-\sqrt{\frac{\gamma d}{1-\gamma}}\frac{a_v}{\|z\|}}^{0} e^{-t^2/2}dt$$
$$\geq \frac{1}{2} - \frac{1}{\sqrt{2\pi}}\left(\sqrt{\frac{\gamma d}{1-\gamma}}\frac{a_v}{\|z\|}\right).$$

Another bound can be obtained using the following inequality (Pólya, 1949, (1.5)):

$$\frac{1}{\sqrt{2\pi}}\int_0^x e^{-t^2/2}dt \le \sqrt{1-e^{-2x^2/\pi}}.$$

We proceed

$$
\begin{aligned}
\mathbb{P}(X \le 0) &= \frac{1}{2} - \frac{1}{\sqrt{2\pi}}\int_{-\sqrt{\frac{\gamma d}{1-\gamma}}\frac{a_v}{\|z\|}}^0 e^{-t^2/2}dt \\
&= \frac{1}{2} - \frac{1}{\sqrt{2\pi}}\int_0^{\sqrt{\frac{\gamma d}{1-\gamma}}\frac{a_v}{\|z\|}} e^{-t^2/2}dt \\
&\ge \frac{1}{2} - \frac{1}{2}\sqrt{1 - \exp\left(-\frac{2\gamma d a_v^2}{\pi(1-\gamma)\|z\|^2}\right)} \\
&\ge \frac{1}{2} - \frac{1}{2}\left(1 - \frac{1}{2}\exp\left(-\frac{2\gamma d a_v^2}{\pi(1-\gamma)\|z\|^2}\right)\right) \\
&\ge \frac{1}{4}\exp\left(-\frac{2\gamma d a_v^2}{\pi(1-\gamma)\|z\|^2}\right). \qquad \square
\end{aligned}
$$

The following result establishes benign and non-benign overfitting for linear models and data as per Definition 2.1, with a phase transition between these outcomes depending on the signal to noise parameter $\gamma$.

**Theorem C.2.** *In the context of the data model described in Section 2, let $w^*$ be a max-margin linear classifier of the training data. Let $\mathcal{A} : \mathbb{R}^{n\times d} \times \{\pm 1\}^n \to \mathbb{R}^d$ be a learning algorithm which is approximately margin-maximizing, Definition 2.3. For $\delta \in (0,1]$, let $d = \Omega\left(n + \log\frac{1}{\delta}\right)$. Then with probability at least $1-\delta$ over the randomness of the training data $(X, \hat{y})$, the following hold with $\epsilon$ denoting the generalization error of $\mathcal{A}(X, \hat{y})$.*

> *(A) If $\gamma = \Omega(|\mathcal{A}|^2 n^{-1})$ and $k = O(|\mathcal{A}|^{-2}n)$, then $\exp\left(-C_1 dk^{-1}\right) \le \epsilon \le \exp\left(-C_2 dk^{-1}|\mathcal{A}|^{-2}\right)$ for fixed positive constants $C_1$ and $C_2$.*

> *(B) If $\gamma = O(|\mathcal{A}|^{-2}d^{-1})$, then $\epsilon \ge \frac{1}{2} - \sqrt{Cd\gamma|\mathcal{A}|^2}$ for a fixed positive constant $C$.*

*Proof.* For training data $(X, \hat{y})$ let $w = \mathcal{A}(X, \hat{y})$ be the learned linear classifier. First, recall $x_i = \sqrt{\gamma}y_i v + \sqrt{1-\gamma}n_i$ for all $i \in [n]$, $\|v\| = 1$, and observe that we can decompose the vector $w$ as $w = a_v v + z$, where $a_v \in \mathbb{R}$, and $z \perp v$. As a result, for each $i \in [n]$,

$$\hat{y}_i\langle x_i, w\rangle = \sqrt{\gamma}a_v\beta_i + \sqrt{1-\gamma}\hat{y}_i\langle n_i, z\rangle. \tag{8}$$

First we establish *(A)*. As $d = \Omega\left(n + \log\frac{1}{\delta}\right)$, Lemmas 4.1 and B.3 show that with probability at least $1 - \frac{\delta}{2}$, $\|N\|^2, \|N_{\mathcal{G}}\|^2, \|N_{\mathcal{B}}\| \le C$ and $\|w^*\| \le C\sqrt{\frac{1}{\gamma} + \frac{k}{1-\gamma}}$. Here $N_{\mathcal{G}}$ and $N_{\mathcal{B}}$ denote the matrices formed by taking only the rows of $N$ which satisfy $\beta = 1$ and $\beta = -1$, respectively. We denote this event $\omega$ and condition on it in all that follows for the proof of *(A)*. As $\mathcal{A}$ is approximately max margin then given equation 8 we have for all $i \in \mathcal{G}$

$$1 \le \sqrt{\gamma}a_v + \sqrt{1-\gamma}\hat{y}_i\langle n_i, z\rangle.$$

Suppose that $\sqrt{\gamma}a_v < \frac{1}{2}$. Then the above inequality implies $\sqrt{1-\gamma}\hat{y}_i\langle n_i, z\rangle \ge \frac{1}{2}$ for all $i \in \mathcal{G}$. Squaring and then summing this expression over all $i \in \mathcal{G}$ it follows that

$$
\begin{aligned}
\frac{n-k}{4} &\le (1-\gamma)\sum_{i\in\mathcal{G}}|\langle n_i, z\rangle|^2 \\
&\le (1-\gamma)\|N_{\mathcal{G}}z\|^2 \\
&\le (1-\gamma)\|N_{\mathcal{G}}\|^2\|z\|^2.
\end{aligned}
$$

Since $\mathcal{A}$ is approximately margin-maximizing,

$$
\begin{aligned}
\frac{n-k}{4} &\leq (1-\gamma)C\|\boldsymbol{z}\|^2 \\
&\leq (1-\gamma)C\|\boldsymbol{w}\|^2 \\
&\leq C|\mathcal{A}|^2(1-\gamma)\|\boldsymbol{w}^*\|^2 \\
&\leq C|\mathcal{A}|^2(1-\gamma)\left(\frac{1}{\gamma} + \frac{k}{1-\gamma}\right) \\
&\leq \frac{C|\mathcal{A}|^2}{\gamma} + C|\mathcal{A}|^2 k,
\end{aligned}
$$

which further implies $n \leq \frac{C|\mathcal{A}|^2}{\gamma} + C|\mathcal{A}|^2 k$ for some other constant $C$. For this inequality to hold, either $\frac{C|\mathcal{A}|^2}{\gamma} \geq \frac{n}{2}$ or $C|\mathcal{A}|^2 k \geq \frac{n}{2}$. With $k \leq \frac{n}{4C|\mathcal{A}|^2}$ and $\gamma \geq \frac{1}{k}$, neither of these can be true and therefore we conclude that $\sqrt{\gamma}a_v \geq \frac{1}{2}$. Then

$$
\begin{aligned}
\frac{a_v^2}{\|\boldsymbol{z}\|^2} &\geq \frac{1}{4\gamma\|\boldsymbol{z}\|^2} \\
&\geq \frac{1}{4\gamma\|\boldsymbol{w}\|^2} \\
&\geq \frac{1}{4\gamma|\mathcal{A}|^2\|\boldsymbol{w}^*\|^2} \\
&\geq \frac{C}{|\mathcal{A}|^2\gamma}\frac{1}{\frac{1}{\gamma} + \frac{k}{1-\gamma}} \\
&\geq \frac{C}{2|\mathcal{A}|^2\gamma}\frac{1-\gamma}{k}
\end{aligned}
$$

for a positive constant $C$. Letting $(\boldsymbol{x}, y)$ denote a test point pair, then by Lemma C.1 it follows that for a different constant $C$

$$
\begin{aligned}
\mathbb{P}(y\langle \boldsymbol{w}, \boldsymbol{x}\rangle \leq 0 \mid \omega) &\leq \exp\left(-\frac{d}{2}\frac{\gamma}{1-\gamma}\frac{a_v^2}{\|\boldsymbol{z}\|^2}\right) \\
&\leq \exp\left(-\frac{Cd}{|\mathcal{A}|^2 k}\right).
\end{aligned}
$$

Hence the generalization error is at most $\epsilon$ when $\omega$ occurs, which happens with probability at least $1-\delta$. This establishes the upper bound of *(A)*.

For the lower bound of *(A)*, since $\boldsymbol{w}$ is a linear classifier,

$$
\langle \boldsymbol{n}_i, \boldsymbol{z}\rangle \geq \frac{1}{\sqrt{1-\gamma}}(1 + a_v\sqrt{\gamma}) \geq a_v\sqrt{\frac{\gamma}{1-\gamma}}
$$

for all $i \in \mathcal{B}$, from which we conclude $|\langle \boldsymbol{n}_i, \boldsymbol{z}\rangle| \geq a_v\sqrt{\gamma}$. This implies

$$
\|\boldsymbol{N}_{\mathcal{B}}\boldsymbol{z}\| \geq a_V\sqrt{\frac{k\gamma}{1-\gamma}}
$$

Along with $\|\boldsymbol{N}_{\mathcal{B}}\boldsymbol{z}\| \leq \|\boldsymbol{N}_{\mathcal{B}}\|\|\boldsymbol{z}\| \leq C\|\boldsymbol{z}\|$ we conclude

$$
\|\boldsymbol{z}\| \geq \frac{a_v}{C}\sqrt{\frac{k\gamma}{1-\gamma}}.
$$

With this bound we then bound

$$
\frac{a_v}{\|\boldsymbol{z}\|} \leq C \cdot \sqrt{\frac{1-\gamma}{k\gamma}}.
$$

By Lemma C.1 we then can bound

$$\mathbb{P}(y\langle \boldsymbol{w}, \boldsymbol{x}\rangle \le 0 \mid \omega) \ge \frac{1}{4}\exp\left(-\frac{6d}{\pi}\frac{\gamma}{1-\gamma}\frac{a_v^2}{\|\boldsymbol{z}\|^2}\right)$$

$$\ge \exp\left(-\frac{Cd}{k}\right)$$

for a new constant $C$, provided $a_v$ is positive. In the last line we can bound $\frac{d}{k}$ below as $d \ge n$ and $k = O(n)$. If $a_v$ is negative then the generalization error is at least $\frac{1}{2}$, which is still bounded below by $\exp(-Cd/k)$.

We now turn our attention to *(B)*. As $d = \Omega(n + \log\frac{1}{\delta})$, from Lemmas 4.1 and B.4 with probability at least $1 - \delta$ it holds that $\|\boldsymbol{N}_{\mathcal{G}}\|^2 \le C$ and $\|\boldsymbol{w}^*\| \le \frac{C\sqrt{n}}{\sqrt{1-\gamma}}$. We denote this event $\omega'$ and condition on it in all that follows for the proof of *(B)*. In particular,

$$a_v^2 + \|\boldsymbol{z}\|^2 = \|\boldsymbol{w}\|^2$$

$$\le |\mathcal{A}|^2\|\boldsymbol{w}^*\|^2$$

$$\le \frac{C|\mathcal{A}|^2 n}{1-\gamma}. \tag{9}$$

For all $i \in [n]$,

$$1 \le \sqrt{\gamma}\beta_i a_v + \sqrt{1-\gamma}\hat{y}_i\langle \boldsymbol{n}_i, \boldsymbol{z}\rangle.$$

For this inequality to hold, either $|\sqrt{\gamma}a_v| \ge 1/2$ or $\sqrt{1-\gamma}\hat{y}_i\langle \boldsymbol{n}_i, \boldsymbol{z}\rangle \ge 1/2$ for all $i \in [n]$. If $|\sqrt{\gamma}a_v| \ge 1/2$, then with $\gamma \le \frac{1}{4(C+1)|\mathcal{A}|^2 d}$ we have

$$a_v^2 \ge 2C|\mathcal{A}|^2 d \ge 2C|\mathcal{A}|^2 n.$$

However, from equation 9 we have

$$2C|\mathcal{A}|^2 n > \frac{C|\mathcal{A}|^2 n}{1-\gamma} \ge a_v^2$$

which is a contradiction. Therefore, under the regime specified there exists a $C > 0$ such that with $\gamma \le \frac{1}{C|\mathcal{A}|^2 d}$, we have $\sqrt{1-\gamma}\hat{y}_i\langle \boldsymbol{n}_i, \boldsymbol{z}\rangle \ge 1/2$ for all $i \in [n]$. Rearranging, squaring and summing over all $i \in [n]$ yields

$$\frac{n}{4(1-\gamma)} \le \sum_{i=1}^n |\langle \boldsymbol{n}_i, \boldsymbol{z}\rangle|^2 \le \|\boldsymbol{N}\boldsymbol{z}\|^2 \le C\|\boldsymbol{z}\|^2,$$

where the final inequality follows from conditioning on $\omega$. Then

$$\frac{a_v^2}{\|\boldsymbol{z}\|^2} \le \frac{\|\boldsymbol{w}\|^2}{\|\boldsymbol{z}\|^2}$$

$$\le \frac{|\mathcal{A}|^2\|\boldsymbol{w}^*\|^2}{\|\boldsymbol{z}\|^2}$$

$$\le \frac{C|\mathcal{A}|^2\frac{n}{1-\gamma}}{\frac{n}{1-\gamma}}$$

$$\le C|\mathcal{A}|^2,$$

where the constant $C > 0$ may vary between inequalities. Letting $(\boldsymbol{x}, y)$ denote a test point pair, by Lemma C.1 it follows that

$$\mathbb{P}(y\langle \boldsymbol{w}, \boldsymbol{x}\rangle \le 0 \mid \omega) \ge \frac{1}{2} - \sqrt{\frac{d\gamma}{2\pi(1-\gamma)}}\frac{a_v}{\|\boldsymbol{z}\|}$$

$$\ge \frac{1}{2} - \sqrt{\frac{Cd\gamma|\mathcal{A}|^2}{1-\gamma}}$$

$$\ge \frac{1}{2} - \sqrt{Cd\gamma|\mathcal{A}|^2}.$$

Hence the generalization error is at least $\frac{1}{2} - \sqrt{Cd\gamma|\mathcal{A}|^2}$ when $\omega'$ occurs, which happens with probability at least $1 - \delta$. This establishes *(B)*. $\square$

# Appendix D   Leaky ReLU Networks

In this section we consider a leaky ReLU network $f : \mathbb{R}^{2m} \times \mathbb{R}^d \to \mathbb{R}$ with forward pass given by

$$f(\boldsymbol{W}, \boldsymbol{x}) = \sum_{j=1}^{2m} (-1)^j \sigma(\langle \boldsymbol{w}_j, \boldsymbol{x}_i \rangle),$$

where $\sigma(z) = \max(\alpha z, z)$ for some $\alpha \in (0, 1)$. For any such network, we may decompose the neuron weights $\boldsymbol{w}_j$ into a signal component and a noise component,

$$\boldsymbol{w}_j = a_j \boldsymbol{v} + \boldsymbol{z}_j,$$

where $a_j \in \mathbb{R}$ and $\boldsymbol{z}_j \in \mathbb{R}^d$ satisfies $\boldsymbol{z}_j \perp \boldsymbol{v}$. The ratio $a_j / \|\boldsymbol{z}_j\|$ therefore grows with the alignment of $\boldsymbol{w}_j$ with the signal and shrinks if $\boldsymbol{w}_j$ instead aligns more with the noise. Collecting the noise components of the weight vectors, let $\boldsymbol{Z} \in \mathbb{R}^{(2m) \times d}$ be the matrix whose $j$-th row is $\boldsymbol{z}_j$. In order to track the alignment of the network as a whole with the signal versus noise subspaces we introduce the following quantities. Let

$$A_1 = f(\boldsymbol{W}, \boldsymbol{v}) = \sum_{j=1}^{2m} (-1)^j \sigma(a_j),$$

$$A_{-1} = f(\boldsymbol{W}, -\boldsymbol{v}) = \sum_{j=1}^{2m} (-1)^{j+1} \sigma(-a_j)$$

be referred to as the positive and negative signal activation of the network respectively. Moreover, define

$$A_{\min} = \min(A_1, A_{-1})$$

as the worst-case signal activation of the network, and

$$A_{\text{lin}} = \sum_{j=1}^{2m} (-1)^j a_j$$

as the linearized network activation. To measure the amount of noise the network learns we define

$$\boldsymbol{z}_{\text{lin}} = \sum_{j=1}^{2m} (-1)^j \boldsymbol{z}_j.$$

## D.1   Training dynamics

**Theorem 3.1.** *Let* $f : \mathbb{R}^p \times \mathbb{R}^n \to \mathbb{R}$ *be a leaky ReLU network with forward pass as defined by equation 1. Suppose the step size* $\eta$ *and initialization condition* $\lambda$ *satisfy Assumption 1. Then for any linearly separable data set* $(\boldsymbol{X}, \hat{\boldsymbol{y}})$ $\mathcal{A}_{GD}(\boldsymbol{X}, \hat{\boldsymbol{y}}, \eta, \lambda)$ *converges after* $T$ *iterations, where*

$$T \leq \frac{C \|\boldsymbol{w}^*\|^2}{\eta \alpha^2 m}.$$

*Furthermore* $\mathcal{A}_{GD}$ *is approximately margin maximizing on* $f$ *(Definition 2.3) with*

$$|\mathcal{A}_{GD}| \leq \frac{C}{\alpha \sqrt{m}}.$$

*Proof.* Our approach is to adapt a classical technique used for the proof of convergence of the Perceptron algorithm for linearly separable data. This is also the approach adopted by Brutzkus et al. (2018). The key idea of the proof is to bound in terms of the number of updates both the norm of the learned vector $\boldsymbol{w}$ as well as its alignment with any linear separator of the data. From the Cauchy-Schwarz inequality these bounds cannot cross, and this in turn bounds the number of updates that can occur. Analogously, we track the alignment of $\boldsymbol{W}^{(t)}$ with the max-margin classifier along with the Frobenius norm of the $\boldsymbol{W}^{(t)}$. To this end denote

$$G(t) = \|\boldsymbol{W}_j^{(t)}\|_F^2$$

and

$$F(t) = \sum_{j=1}^{2m} (-1)^j \langle \boldsymbol{w}_j^{(t)}, \boldsymbol{w}^* \rangle,$$

where $\boldsymbol{w}^*$ is a max-margin linear classifier of the dataset. Recall that $\mathcal{F}^{(t)} = \{i \in [n] : \hat{y}_i f(\boldsymbol{W}^{(t)}, \boldsymbol{x}_i) < 1\}$ denotes the number of active data points at training step $t$. We also define $U(t) = \sum_{s=0}^{t-1} |\mathcal{F}^{(s)}|$ to be the number of data point updates between iterations $0$ and $t$. First, by Cauchy-Schwarz

$$1 \leq \langle \boldsymbol{w}^*, \boldsymbol{x}_i \rangle \leq \|\boldsymbol{w}^*\| \cdot \|\boldsymbol{x}_i\|$$

for all $i \in [n]$. Therefore,

$$\|\boldsymbol{w}^*\| \geq \frac{1}{\min_{i \in [n]} \|\boldsymbol{x}_i\|}.$$

By Assumption 1, for all $j \in [2m]$,

$$\|\boldsymbol{w}_j^{(0)}\| \leq \frac{\sqrt{\alpha}}{m \min_{i \in [n]} \|\boldsymbol{x}_i\|} \leq \frac{\|\boldsymbol{w}^*\|}{\alpha m}. \tag{10}$$

For all $t \geq 0$, the update rule of GD implies

$$G(t+1) = \sum_{j=1}^{2m} \|\boldsymbol{w}_j^{(t+1)}\|^2$$

$$= \sum_{j=1}^{2m} \left\| \boldsymbol{w}_j^{(t)} + \eta(-1)^j \sum_{i \in \mathcal{F}^{(t)}} \dot{\sigma}(\langle \boldsymbol{w}_j^{(t)}, \boldsymbol{x}_i \rangle) \hat{y}_i \boldsymbol{x}_i \right\|^2$$

$$= \sum_{j=1}^{2m} \|\boldsymbol{w}_j^{(t)}\|^2 + 2\eta \sum_{j=1}^{2m} \sum_{i \in \mathcal{F}^{(t)}} (-1)^j \dot{\sigma}(\langle \boldsymbol{w}_j^{(t)}, \boldsymbol{x}_i \rangle) \hat{y}_i \langle \boldsymbol{w}_j^{(t)}, \boldsymbol{x}_i \rangle + \eta^2 \sum_{j=1}^{2m} \sum_{i,l \in \mathcal{F}^{(t)}} \dot{\sigma}(\langle \boldsymbol{w}_j^{(t)}, \boldsymbol{x}_i \rangle) \langle \hat{y}_i \boldsymbol{x}_i, \hat{y}_i \boldsymbol{x}_\ell \rangle$$

$$\leq \sum_{j=1}^{2m} \|\boldsymbol{w}_j^{(t)}\|^2 + 2\eta \sum_{j=1}^{2m} \sum_{i \in \mathcal{F}^{(t)}} (-1)^j \dot{\sigma}(\langle \boldsymbol{w}_j^{(t)}, \boldsymbol{x}_i \rangle) \hat{y}_i \langle \boldsymbol{w}_j^{(t)}, \boldsymbol{x}_i \rangle + 2m\eta^2 |\mathcal{F}^{(t)}|^2 \max_{i \in [n]} \|\boldsymbol{x}_i\|^2.$$

Observe that for all $z \in \mathbb{R}$, $\sigma(s) = \dot{\sigma}(z)z$, so can rewrite the second term of the above expression as

$$2\eta \sum_{j=1}^{2m} \sum_{i \in \mathcal{F}^{(t)}} (-1)^j \dot{\sigma}(\langle \boldsymbol{w}_j^{(t)}, \boldsymbol{x}_i \rangle) \hat{y}_i \langle \boldsymbol{w}_j^{(t)}, \boldsymbol{x}_i \rangle = 2\eta \sum_{j=1}^{2m} \sum_{i \in \mathcal{F}^{(t)}} (-1)^j \sigma(\langle \boldsymbol{w}_j^{(t)}, \boldsymbol{x}_i \rangle) \hat{y}_i$$

$$= 2\eta \sum_{i \in \mathcal{F}^{(t)}} \hat{y}_i f(\boldsymbol{W}^{(t)}, \boldsymbol{x}_i)$$

$$< 2\eta \sum_{i \in \mathcal{F}^{(t)}} 1$$

$$= 2\eta |\mathcal{F}^{(t)}|$$

where the inequality in the second-to-last line follows as we are summing over $\mathcal{F}^{(t)}$, which by definition consists of the $i \in [n]$ such that $\hat{y}_i f(\boldsymbol{W}^{(t)}, \boldsymbol{x}_i) < 1$. As a result we obtain

$$G(t+1) \leq \sum_{j=1}^{2m} \|\boldsymbol{w}_j^{(t)}\|^2 + 2\eta |\mathcal{F}^{(t)}| + 2m\eta^2 |\mathcal{F}^{(t)}|^2 \max_{i \in [n]} \|\boldsymbol{x}_i\|^2$$

$$= G(t) + 2\eta |\mathcal{F}^{(t)}| + 2m\eta^2 |\mathcal{F}^{(t)}|^2 \max_{i \in [n]} \|\boldsymbol{x}_i\|^2$$

$$\leq G(t) + 4\eta |\mathcal{F}^{(t)}|,$$

where the last line follows since

$$\eta \leq \frac{1}{mn \max_{i \in [n]} \|\boldsymbol{x}_i\|^2} \leq \frac{1}{|\mathcal{F}^{(t)}| m \max_{i \in [n]} \|\boldsymbol{x}_i\|^2}.$$

By equation 10, the initialization satisfies

$$G(0) = \sum_{j=1}^{2m} \|\boldsymbol{w}_j^{(0)}\|^2$$

$$\leq \sum_{j=1}^{2m} \frac{\|\boldsymbol{w}^*\|^2}{\alpha^2 m^2}$$

$$= \frac{2\|\boldsymbol{w}^*\|^2}{\alpha^2 m}$$

So by induction, for all $t \geq 0$

$$G(t) \leq \frac{2\|\boldsymbol{w}^*\|^2}{\alpha^2 m} + 3\eta \sum_{s=0}^{t-1} |\mathcal{F}^{(s)}| = \frac{2\|\boldsymbol{w}^*\|^2}{\alpha^2 m} + 3\eta U(t). \tag{11}$$

Next we find a bound for $F(t)$. For all $t \geq 0$ then by definition of the GD update

$$F(t+1) = \sum_{j=1}^{2m} (-1)^j \langle \boldsymbol{w}_j^{(t+1)}, \boldsymbol{w}^* \rangle$$

$$= \sum_{j=1}^{2m} (-1)^j \langle \boldsymbol{w}_j^{(t)}, \boldsymbol{w}^* \rangle + \eta \sum_{j=1}^{2m} \sum_{i \in \mathcal{F}^{(t)}} \dot{\sigma}(\langle \boldsymbol{w}_j^{(t)}, \boldsymbol{x}_i \rangle) \hat{y}_i \langle \boldsymbol{w}^*, \boldsymbol{x}_i \rangle.$$

Since $\hat{y}_i \langle \boldsymbol{w}^*, \boldsymbol{x}_i \rangle \geq 1$ for all $i \in [n]$, the above expression is bounded below by

$$\sum_{j=1}^{2m} (-1)^j \langle \boldsymbol{w}_j^{(t)}, \boldsymbol{w}^* \rangle + \eta \sum_{j=1}^{2m} \sum_{i \in \mathcal{F}^{(t)}} \dot{\sigma}(\langle \boldsymbol{w}_j^{(t)}, \boldsymbol{x}_i \rangle) \hat{y}_i = F(t) + \eta \sum_{j=1}^{2m} \sum_{i \in \mathcal{F}^{(t)}} \dot{\sigma}(\langle \boldsymbol{w}_j^{(t)}, \boldsymbol{x}_i \rangle)$$

$$\geq F(t) + \eta \sum_{j=1}^{2m} \sum_{i \in \mathcal{F}^{(t)}} \alpha$$

$$\geq F(t) + 2\eta m \alpha |\mathcal{F}^{(t)}|.$$

Hence unrolling the update for GD for all $t \geq 0$ it follows that

$$F(t+1) \geq F(0) + 2\eta m \alpha \sum_{s=0}^{t-1} |\mathcal{F}^{(s)}|.$$

At initialization, by equation 10 then

$$F(0) = \sum_{j=1}^{2m} (-1)^j \langle \boldsymbol{w}_j^{(0)}, \boldsymbol{w}^* \rangle$$

$$\geq -\sum_{j=1}^{2m} \|\boldsymbol{w}_j^{(0)}\| \cdot \|\boldsymbol{w}^*\|$$

$$\geq -\sum_{j=1}^{2m} \frac{\|\boldsymbol{w}^*\|^2}{\alpha m}$$

$$= -\frac{2\|\boldsymbol{w}^*\|^2}{\alpha}.$$

Therefore by induction, for all $t \geq 0$ we have

$$F(t) \geq -\frac{2\|\boldsymbol{w}^*\|^2}{\alpha} + 2\eta m \alpha \sum_{s=0}^{t-1} |\mathcal{F}^{(s)}|$$

$$= -\frac{2\|\boldsymbol{w}^*\|^2}{\alpha} + 2\eta m \alpha U(t).$$

Combining our bounds for $F(t)$ and $G(t)$, we obtain

$$-\frac{2\|\boldsymbol{w}^*\|^2}{\alpha} + 2\eta m\alpha U(t) \leq F(t)$$

$$= \sum_{j=1}^{2m}(-1)^j\langle \boldsymbol{w}_j^{(t)}, \boldsymbol{w}^*\rangle$$

$$\leq \|\boldsymbol{w}^*\|\sum_{j=1}^{2m}\|\boldsymbol{w}_j^{(t)}\|$$

$$\leq \|\boldsymbol{w}^*\|\left(2m\sum_{j=1}^{2m}\|\boldsymbol{w}_j^{(t)}\|^2\right)^{1/2}$$

$$= \|\boldsymbol{w}^*\|(2mG(t))^{1/2}$$

$$\leq \|\boldsymbol{w}^*\|\left(\frac{4\|\boldsymbol{w}^*\|^2}{\alpha^2} + 6m\eta U(t)\right)^{1/2}.$$

This implies that either

$$-\frac{2\|\boldsymbol{w}^*\|^2}{\alpha} + 2\eta m\alpha U(t) \leq 0 \tag{12}$$

or

$$\left(-\frac{2\|\boldsymbol{w}^*\|^2}{\alpha} + 2\eta m\alpha U(t)\right)^2 \leq \|\boldsymbol{w}^*\|^2\left(\frac{4\|\boldsymbol{w}^*\|^2}{\alpha^2} + 6m\eta U(t)\right). \tag{13}$$

If (12) holds, then

$$U(t) \leq \frac{\|\boldsymbol{w}^*\|^2}{\eta\alpha^2 m}.$$

If (13) holds, then rearranging yields

$$4\eta^2 m^2\alpha^2 U(t)^2 \leq 14\|\boldsymbol{w}^*\|^2\eta m U(t)$$

$$U(t) \leq \frac{7\|\boldsymbol{w}^*\|^2}{2\eta\alpha^2 m}.$$

Therefore, in both cases there exists a constant $C$ such that

$$U(t) \leq \frac{C\|\boldsymbol{w}^*\|^2}{\eta\alpha^2 m}. \tag{14}$$

This holds for all $t \in \mathbb{N}$ and therefore

$$\sum_{t=0}^{\infty}|\mathcal{F}^{(t)}| \leq \frac{C\|\boldsymbol{w}^*\|^2}{\eta\alpha^2 m} < \infty.$$

This implies that there exists $s \in \mathbb{N}$ such that $|\mathcal{F}^{(s)}| = 0$. Let $T \in \mathbb{N}$ be the minimal iteration such that $|\mathcal{F}^{(T)}| = 0$. Then for all $i \in [n]$ $\hat{y}_i f(\boldsymbol{W}^{(T)}, \boldsymbol{x}_i) \geq 1$. So the network achieves zero loss and also has zero gradient at iteration $T$. In particular,

$$T = \sum_{t=0}^{T-1}1 \leq \sum_{t=0}^{T-1}|\mathcal{F}^{(t)}| \leq \frac{C\|\boldsymbol{w}^*\|^2}{\eta\alpha^2 m}.$$

To bound $|\mathcal{A}_{GD}|$ we combine equations (14) and (11) to obtain

$$G(T) \leq \frac{2\|\boldsymbol{w}^*\|^2}{\alpha^2 m} + 3\eta U(t)$$

$$\leq \frac{2\|\boldsymbol{w}^*\|^2}{\alpha^2 m} + \frac{C\|\boldsymbol{w}^*\|^2}{\eta\alpha^2 m}$$

$$\leq \frac{C\|\boldsymbol{w}^*\|^2}{\alpha^2 m}.$$

As a result for all linearly separable datasets $(\boldsymbol{X}, \hat{\boldsymbol{y}})$

$$\frac{\|\boldsymbol{W}\|_F}{\|\boldsymbol{w}^*\|} = \frac{C}{\alpha\sqrt{m}}$$

and therefore

$$|\mathcal{A}_{GD}| \leq \frac{C}{\alpha\sqrt{m}}$$

as claimed. □

The training dynamics of gradient descent also give us the following result relating the linearization of the noise component of the network to the noise component of the network itself.

**Lemma D.1.** *Let* $\lambda, \delta > 0$. *Suppose that* $d \geq \Omega\left(n + \log\frac{1}{\delta}\right)$. *In the context of training data* $(\boldsymbol{X}, \hat{\boldsymbol{y}})$ *sampled under the data model given in Definition 2.1, let* $\boldsymbol{W} = \mathcal{A}_{GD}(\boldsymbol{X}, \hat{\boldsymbol{y}}, \eta, \lambda)$. *Then with probability at least* $1 - \delta$ *over the randomness of* $(\boldsymbol{X}, \hat{\boldsymbol{y}})$

$$\|\boldsymbol{Z}\|_F^2 - 2\lambda\sqrt{2m}\|\boldsymbol{Z}\|_F - 2m\lambda^2 \leq \frac{C}{\alpha m}\left(\|\boldsymbol{z}_{\text{lin}}\| + 2m\lambda\right)^2.$$

*Proof.* At each iteration of gradient descent,

$$\boldsymbol{w}_j^{(t+1)} = \boldsymbol{w}_j^{(t)} + \eta(-1)^j \sum_{i=1}^n b_{ij}^{(t)} \hat{y}_i \boldsymbol{x}_i,$$

where

$$b_{ij}^{(t)} = \begin{cases} 0 & \text{if } \hat{y}_i f(\boldsymbol{W}^{(t)}, \boldsymbol{x}_i) \geq 1 \\ 1 & \text{if } \hat{y}_i f(\boldsymbol{W}^{(t)}, \boldsymbol{x}_i) < 1 \text{ and } \langle \boldsymbol{w}_j^{(t)}, \boldsymbol{x}_i \rangle \geq 0 \ . \\ \alpha & \text{otherwise.} \end{cases}$$

Let $T$ be the iteration at which gradient descent terminates. Then for each $j \in [2m]$,

$$\boldsymbol{w}_j = \boldsymbol{w}_j^{(T)} = \boldsymbol{w}_j^{(0)} + \eta(-1)^j \sum_{t=0}^{T-1}\sum_{i=1}^n b_{ij}^{(t)} \hat{y}_i \boldsymbol{x}_i.$$

Then the noise component of $\boldsymbol{w}_j$ is given by

$$\begin{aligned}
\boldsymbol{z}_j &= \boldsymbol{w}_j - \langle \boldsymbol{w}_j, \boldsymbol{v} \rangle \boldsymbol{v} \\
&= \boldsymbol{w}_j^{(0)} - \langle \boldsymbol{w}_j^{(0)}, \boldsymbol{v} \rangle \boldsymbol{v} + \eta(-1)^j \sum_{t=0}^{T-1}\sum_{i=1}^n b_{ij}^{(t)} \hat{y}_i (\boldsymbol{x}_i - \langle \boldsymbol{x}_i, \boldsymbol{v} \rangle \boldsymbol{v}) \\
&= \boldsymbol{w}_j^{(0)} - \langle \boldsymbol{w}_j^{(0)}, \boldsymbol{v} \rangle \boldsymbol{v} + \eta(-1)^j \sum_{t=0}^{T-1}\sum_{i=1}^n b_{ij}^{(t)} \hat{y}_i \boldsymbol{n}_i.
\end{aligned}$$

Define

$$\hat{\boldsymbol{z}}_j = \boldsymbol{z}_j - \boldsymbol{w}_j^{(0)} + \langle \boldsymbol{w}_j^{(0)}, \boldsymbol{v} \rangle \boldsymbol{v}$$

and let

$$\hat{\boldsymbol{z}}_{\text{lin}} = \sum_{j=1}^{2m} (-1)^j \hat{\boldsymbol{z}}_j,$$

Then for all $j \in [2n]$,

$$\begin{aligned}
(\|\hat{\boldsymbol{z}}_j\| - \|\boldsymbol{z}_j\|)^2 &\leq \|\hat{\boldsymbol{z}}_j - \boldsymbol{z}_j\|^2 \\
&= \|\boldsymbol{w}_j^{(0)} - \langle \boldsymbol{w}_j^{(0)}, \boldsymbol{v} \rangle \boldsymbol{v}\|^2 \\
&\leq \|\boldsymbol{w}_j^{(0)}\|^2 \\
&\leq \lambda^2.
\end{aligned}$$

Furthermore, if $\|z_j\| \leq \|\hat{z}_j\|$ then the above implies $\|\hat{z}_j\| \leq \|z_j\| + \lambda$ while if $\|z_j\| \geq \|\hat{z}_j\|$ then this inequality holds trivially. As a result,

$$
\begin{aligned}
\left| \|\hat{z}_j\|^2 - \|z_j\|^2 \right| &= |(\|\hat{z}_j\| + \|z_j\|) \cdot (\|\hat{z}_j\| - \|z_j\|)| \\
&\leq |\|\hat{z}_j\| + \|z_j\|| \cdot |\|\hat{z}_j\| - \|z_j\||) \\
&\leq (2\|z_j\| + \lambda)(\lambda).
\end{aligned}
$$

If $\|z_j\| \geq \|\hat{z}_j\|$ then the above implies $\|\hat{z}_j\|^2 \geq \|z_j\|^2 - \lambda(2\|z_j\| + \lambda)$, if $\|z_j\| \leq \|\hat{z}_j\|$ this inequality is trivially true. As a result,

$$
\begin{aligned}
\sum_{j=1}^{2m} \|\hat{z}_j\|^2 &\geq \sum_{j=1}^{2m} (\|z_j\|^2 - \lambda(2\|z_j\| + \lambda)) \\
&= \sum_{j=1}^{2m} \|z_j\|^2 - 2\lambda \sum_{j=1}^{2m} \|z_j\| - 2m\lambda^2 \\
&\geq \sum_{j=1}^{2m} \|z_j\|^2 - 2\lambda\sqrt{2m} \left( \sum_{j=1}^{2m} \|z_j\|^2 \right)^{1/2} - 2m\lambda^2 \\
&= \|Z\|_F^2 - 2\lambda\sqrt{2m}\|Z\|_F - 2m\lambda^2,
\end{aligned} \tag{15}
$$

where the third line is an application of Cauchy-Schwarz. Moreover,

$$
\begin{aligned}
\|\hat{z}_{\mathrm{lin}} - z_{\mathrm{lin}}\| &= \left\| \sum_{j=1}^{2m} (-1)^j (\hat{z}_j - z_j) \right\| \\
&\leq \sum_{j=1}^{2m} \|\hat{z}_j - z_j\| \\
&\leq \sum_{j=1}^{2m} \lambda \\
&\leq 2m\lambda,
\end{aligned}
$$

so

$$
\|\hat{z}_{\mathrm{lin}}\| \geq \|z_{\mathrm{lin}}\| - 2m\lambda. \tag{16}
$$

Let $N' \in \mathbb{R}^{d \times n}$ to be the matrix whose $i$-th column is $\hat{y}_i n_i$, equivalently $N' = N\mathrm{diag}(\hat{y})$. Then

$$
\hat{z}_j = \eta(-1)^j N' c_j,
$$

where $c_j \in \mathbb{R}^n$ is given by

$$
(c_j)_i = \sum_{t=0}^{T-1} b_{ij}^{(t)}.
$$

Due to symmetry of the noise distribution then the columns of $N'$ are i.i.d. with distribution $\mathcal{N}(0_d, d^{-1}(I_d - vv^T))$. Therefore by Lemma 4.1 (and the assumptions $d = \Omega\left(n + \log \frac{1}{\delta}\right)$), with probability at least $1 - \delta$ over the randomness of the training data there exist positive constants $C', C$ such that $C' \leq \sigma_{\min}(N') \leq \sigma_{\max}(N') \leq C$. As a result

$$
C'\eta\|c_j\| \leq \|\hat{z}_j\| \leq C\eta\|c_j\|. \tag{17}
$$

We claim that for any $j, j' \in [2m]$ and $i \in [n]$, $(c_j)_i \geq \alpha(c_{j'})_i$. Indeed, if $\hat{y}_i f(W^{(t)}, x_i) \geq 1$, then $b_{ij}^{(t)} = b_{ij'}^{(t)} = 0$, and if $\hat{y}_i f(W^{(t)}, x_i) < 1$, then both $b_{ij}^{(t)}$ and $b_{ij'}^{(t)}$ are elements of $\{\alpha, 1\}$. This in particular implies that

$$
\langle c_j, c_{j'} \rangle \geq \alpha \langle c_j, c_j \rangle.
$$

Let us define
$$\boldsymbol{c}_{\text{lin}} = \sum_{j=1}^{2m} \boldsymbol{c}_j.$$

Then
$$\|\hat{\boldsymbol{z}}_{\text{lin}}\|^2 = \left\| \sum_{j=1}^{2m} (-1)^j \hat{\boldsymbol{z}}_j \right\|^2$$
$$= \left\| \sum_{j=1}^{2m} \eta \boldsymbol{N}' \boldsymbol{c}_j \right\|^2$$
$$\leq C\eta^2 \left\| \sum_{j=1}^{2m} \boldsymbol{c}_j \right\|^2$$
$$= C\eta^2 \|\boldsymbol{c}_{\text{lin}}\|^2, \tag{18}$$

where we used that $\|\boldsymbol{N}'\| \leq C$ in the third line. We also have

$$\|\boldsymbol{c}_{\text{lin}}\|^2 = \sum_{j=1}^{2m} \sum_{j'=1}^{2m} \langle \boldsymbol{c}_j, \boldsymbol{c}_{j'} \rangle$$
$$\geq \alpha \sum_{j=1}^{2m} \sum_{j'=1}^{2m} \langle \boldsymbol{c}_j, \boldsymbol{c}_j \rangle$$
$$= 2\alpha m \sum_{j=1}^{2m} \|\boldsymbol{c}_j\|^2. \tag{19}$$

Finally we combine our bounds for $\boldsymbol{c}$, $\boldsymbol{z}$, and $\hat{\boldsymbol{z}}$:

$$\|\boldsymbol{Z}\|_F^2 - 2\lambda\sqrt{2m}\|\boldsymbol{Z}\|_F - 2m\lambda^2 \leq \sum_{j=1}^{2m} \|\hat{\boldsymbol{z}}_j\|^2$$
$$\leq C\eta^2 \sum_{j=1}^{2m} \|\boldsymbol{c}_j\|^2$$
$$\leq \frac{C\eta^2}{\alpha m} \|\boldsymbol{c}_{\text{lin}}\|^2$$
$$\leq \frac{C}{\alpha m} \|\hat{\boldsymbol{z}}_{\text{lin}}\|^2$$
$$\leq \frac{C}{\alpha m} \left( \|\boldsymbol{z}_{\text{lin}}\| + 2m\lambda \right)^2.$$

Here we applied equations (15) in the first line, (17) in the second line, (19) in the third line, (18) in the fourth line, and (16) in the fifth line. This establishes both the bounds claimed. $\qquad\square$

## D.2 Benign overfitting

To establish benign overfitting in leaky ReLU networks, we first determine an upper bound on the generalization error of the model in terms of the signal-to-noise ratio of the network weights.

**Lemma D.2.** *Let $\epsilon \in (0,1)$. Suppose that*

$$\frac{A_{\min}}{\|\boldsymbol{Z}\|_F} \geq C_2 \sqrt{\frac{(1-\gamma)m\log\frac{1}{\epsilon}}{\gamma d}}.$$

*Then for test data $(\boldsymbol{x}, y)$ as per Definition 2.1,*

$$\mathbb{P}(yf(\boldsymbol{W}, \boldsymbol{x}) \leq 0) \leq \epsilon.$$

*Proof.* Recall that a test point $(\boldsymbol{x}, y)$ satisfies

$$\boldsymbol{x} = y(\sqrt{\gamma}\boldsymbol{v} + \sqrt{1-\gamma}\boldsymbol{n}),$$

where $\boldsymbol{n} \sim \mathcal{N}(\boldsymbol{0}_d, \frac{1}{d}(\boldsymbol{I}_d - \boldsymbol{v}\boldsymbol{v}^T))$. If $yf(\boldsymbol{W}, \boldsymbol{x}) \leq 0$, then

$$0 \geq yf(\boldsymbol{W}, \boldsymbol{x})$$

$$= \sum_{j=1}^{2m}(-1)^j y\sigma(\langle \boldsymbol{w}_j, \boldsymbol{x} \rangle)$$

$$= \sum_{j=1}^{2m}(-1)^j y\sigma(\langle a_j\boldsymbol{v} + \boldsymbol{z}_j, y(\sqrt{\gamma}\boldsymbol{v} + \sqrt{1-\gamma}\boldsymbol{n}))$$

$$= \sum_{j=1}^{2m}(-1)^j y\sigma(y(\sqrt{\gamma}a_j + \sqrt{1-\gamma}\langle \boldsymbol{z}_j, \boldsymbol{n}\rangle))$$

$$\geq \sum_{j=1}^{2m}(-1)^j y\sigma(y\sqrt{\gamma}a_j) - \sum_{j=1}^{2m}\sqrt{1-\gamma}|\langle \boldsymbol{z}_j, \boldsymbol{n}\rangle|$$

$$= \sqrt{\gamma}A_y - \sum_{j=1}^{2m}\sqrt{1-\gamma}|\langle \boldsymbol{z}_j, \boldsymbol{n}\rangle|.$$

When $A_{\min} \geq 0$, this implies that

$$\gamma A_{\min}^2 \leq (1-\gamma)\left(\sum_{j=1}^{2m}|\langle \boldsymbol{z}_j, \boldsymbol{n}\rangle|\right)^2$$

$$\leq 2m(1-\gamma)\sum_{j=1}^{2m}|\langle \boldsymbol{z}_j, \boldsymbol{n}\rangle|^2$$

$$= 2m(1-\gamma)\|\boldsymbol{Z}\boldsymbol{n}\|^2$$

$$\leq 2m(1-\gamma)\|\boldsymbol{Z}\|_F^2\|\boldsymbol{n}\|^2,$$

where the second inequality is an application of Cauchy-Schwarz. So

$$\mathbb{P}(yf(\boldsymbol{W}, \boldsymbol{x}) \leq 0) \leq \mathbb{P}\left(\|\boldsymbol{Z}\boldsymbol{n}\|^2 \geq \frac{\gamma A_{\min}^2}{2m(1-\gamma)}\right).$$

By Lemma A.1, the above probability is less than $\epsilon$ if

$$\sqrt{\frac{\gamma}{2m(1-\gamma)}}A_{\min} \geq C\|\boldsymbol{Z}\|_F\sqrt{\frac{1}{d}\log\frac{1}{\epsilon}},$$

or equivalently,

$$\frac{A_{\min}}{\|\boldsymbol{Z}\|_F} \geq C_2\sqrt{\frac{(1-\gamma)m\log\frac{1}{\epsilon}}{\gamma d}}.$$

$\square$

We will also need the number of positive labels to be (mildly) balanced with the number of negative labels.

**Lemma D.3.** *Let $\delta > 0$ and suppose that $\ell = \Omega\left(\log\frac{1}{\delta}\right)$. Let $\mathcal{I} \subseteq [n]$ be an arbitrary subset such that $|\mathcal{I}| = \ell$. Consider training data $(\boldsymbol{X}, \boldsymbol{y})$ as per the data model given in Definition 2.1. Then with probability at least $1 - \delta$,*

$$\frac{\ell}{4} \leq |\{i \in \mathcal{S} : y_i = 1\}| \leq \frac{3\ell}{4}.$$

*Proof.* For $i \in \mathcal{I}$ let $Y_i$ be a random variable taking the value 1 if $y_i = 1$ and 0 if $y_i = -1$. Then the $Y_i$ are i.i.d. Bernoulli random variables with $\mathbb{P}(Y_i = 1) = \frac{1}{2}$. Let

$$Y = \sum_{i \in \mathcal{I}} Y_i = |\{i \in \mathcal{S} : y_i = 1\}|$$

so that $\mathbb{E}[Y] = \frac{l}{2}$. By Chernoff's inequality, for all $t \in (0, 1)$,

$$\mathbb{P}\left(\left|Y - \frac{\ell}{2}\right| \geq t\frac{\ell}{2}\right) \leq 2e^{-C\ell t^2}.$$

Setting $t = \frac{1}{2}$, we see that $\frac{\ell}{4} \leq Y \leq \frac{3\ell}{4}$ with probability at least

$$1 - 2\exp\left(-\frac{C\ell}{4}\right) \geq 1 - \delta$$

when $\ell = \Omega\left(\log\frac{1}{\delta}\right)$. $\qquad\square$

We are now able to prove our main benign overfitting result for leaky ReLU networks.

**Theorem 3.2.** *Under the setting given in Assumption 2, let $\delta \in (0, 1)$ and suppose $\mathcal{A}$ is approximately margin-maximizing (Definition 2.3). If $n = \Omega\left(\log\frac{1}{\delta}\right)$, $d = \Omega(n)$, $k = O(\frac{n}{1+m|\mathcal{A}|^2})$, and $\gamma = \Omega\left(\frac{1}{k}\right)$ then there is a fixed positive constant $C$ such that with probability at least $1 - \delta$ over $(\boldsymbol{X}, \hat{\boldsymbol{y}})$*

$$\mathbb{P}(yf(\boldsymbol{W}, \boldsymbol{x}) \leq 0 \mid \boldsymbol{X}, \hat{\boldsymbol{y}}) \leq \exp\left(-C \cdot \frac{d}{k(1+m|\mathcal{A}|^2)}\right).$$

*Proof.* Since $d = \Omega(n) = \Omega\left(n + \log\frac{1}{\delta}\right)$, by Lemma B.3, with probability at least $1 - \frac{\delta}{3}$ over the randomness of the data, the max-margin classifier $\boldsymbol{w}^*$ satisfies

$$\|\boldsymbol{w}^*\| \leq C\sqrt{\frac{1}{\gamma} + \frac{k}{1-\gamma}}.$$

We denote this event by $\omega_1$. For $s \in \{1, -1\}$, let $\mathcal{G}_s$ denote the set of $i \in \mathcal{G}$ such that $\langle \boldsymbol{v}, \boldsymbol{x}_i \rangle = s$. If $n = \Omega\left(\frac{1}{\delta}\right)$ and $k = O(n)$, then $|\mathcal{G}| = \Omega\left(\log\frac{1}{\delta}\right)$. Under these assumptions, by Lemma D.3,

$$|\mathcal{G}_s| \geq \frac{1}{4}|\mathcal{G}| \geq Cn$$

for both $s \in \{1, -1\}$ with probability at least $1 - \frac{\delta}{3}$. We denote this event by $\omega_2$. For $s \in \{1, -1\}$, let $\boldsymbol{N}_{\mathcal{G}_s} \in \mathbb{R}^{|\mathcal{G}_s| \times d}$ be the matrix whose rows are indexed by $\mathcal{G}_s$ and are given by the vectors $\boldsymbol{n}_i$ for $i \in \mathcal{G}_s$. As $d = \Omega(n) = \Omega\left(n + \log\frac{1}{\delta}\right)$ and the rows of $\boldsymbol{N}_{\mathcal{G}_s}$ are drawn mutually i.i.d. from $\mathcal{N}(\boldsymbol{0}_d, d^{-1}(\boldsymbol{I}_d - \boldsymbol{v}^T))$, the following holds by Lemma 4.1. With probability at least $1 - \frac{\delta}{3}$ over the randomness of the training data, $\|\boldsymbol{N}_{\mathcal{G}_s}\| \leq C$ for both $s \in \{1, -1\}$. We denote this event by $\omega_3$. Let $\omega = \omega_1 \cap \omega_2 \cap \omega_3$. By the union bound $\mathbb{P}(\omega) \geq 1 - \delta$. We condition on $\omega$ for the remainder of this proof.

Since $\boldsymbol{W} = \mathcal{A}(\boldsymbol{X}, \hat{\boldsymbol{y}})$ and $\mathcal{A}$ is approximately margin maximizing,

$$\|\boldsymbol{W}\|_F \leq |\mathcal{A}| \cdot \|\boldsymbol{w}^*\|$$

$$\leq C|\mathcal{A}|\sqrt{\frac{1}{\gamma} + \frac{k}{1-\gamma}}. \tag{20}$$

Let $s \in \{-1, 1\}$ be such that $A_s = A_{\min}$. Since the network attains zero loss, for all $i \in \mathcal{G}_s$,

$$1 \le \hat{y}_i f(\boldsymbol{W}, \boldsymbol{x}_i)$$

$$= \sum_{j=1}^{2m} (-1)^j \hat{y}_i \sigma(\langle \boldsymbol{w}_j, \boldsymbol{x}_i \rangle)$$

$$= \sum_{j=1}^{2m} (-1)^j y_i \sigma(\langle a_j \boldsymbol{v} + \boldsymbol{z}_j, \sqrt{\gamma} y_i \boldsymbol{v} + \sqrt{1-\gamma} \boldsymbol{n}_i \rangle)$$

$$= \sum_{j=1}^{2m} (-1)^j y_i \sigma(\sqrt{\gamma} a_j y_i + \sqrt{1-\gamma} \langle \boldsymbol{z}_j, \boldsymbol{n}_i \rangle)$$

$$\le \sum_{j=1}^{2m} (-1)^j y_i \sigma(\sqrt{\gamma} a_j y_i) + \sum_{j=1}^{2m} |\sqrt{1-\gamma} \langle \boldsymbol{z}_j, \boldsymbol{n}_i \rangle|$$

$$= \sqrt{\gamma} \sum_{j=1}^{2m} (-1)^j s \sigma(s a_j) + \sqrt{1-\gamma} \sum_{j=1}^{2m} |\langle \boldsymbol{z}_j, \boldsymbol{n}_i \rangle|$$

$$= \sqrt{\gamma} A_s + \sqrt{1-\gamma} \sum_{j=1}^{2m} |\langle \boldsymbol{z}_j, \boldsymbol{n}_i \rangle|$$

$$= \sqrt{\gamma} A_{\min} + \sqrt{1-\gamma} \sum_{j=1}^{2m} |\langle \boldsymbol{z}_j, \boldsymbol{n}_i \rangle|.$$

Hence, we have either $\sqrt{\gamma} A_s \ge \frac{1}{2}$ or $\sqrt{1-\gamma} \sum_{j=1}^{2m} |\langle \boldsymbol{z}_j, \boldsymbol{n}_i \rangle| \ge \frac{1}{2}$ for all $i \in \mathcal{G}_s$. We consider these two cases separately.

If $\sqrt{\gamma} A_{\min} \ge \frac{1}{2}$, then

$$\frac{A_{\min}}{\|\boldsymbol{Z}\|_F} \ge \frac{A_{\min}}{\|\boldsymbol{W}\|_F}$$

$$\ge \frac{1}{2\sqrt{\gamma} \|\boldsymbol{W}\|_F}$$

$$\ge C \frac{1}{\sqrt{\gamma} |\mathcal{A}| \sqrt{\frac{1}{\gamma} + \frac{k}{1-\gamma}}}$$

$$= C \frac{1}{|\mathcal{A}| \sqrt{1 + \frac{k\gamma}{1-\gamma}}}$$

$$\ge C \frac{1}{|\mathcal{A}| + |\mathcal{A}| \sqrt{\frac{k\gamma}{1-\gamma}}}.$$

Then by Lemma D.2, the network has generalization error less than $\epsilon$ when

$$\frac{1}{|\mathcal{A}| + |\mathcal{A}| \sqrt{\frac{k\gamma}{1-\gamma}}} \ge C \sqrt{\frac{(1-\gamma) m \log \frac{1}{\epsilon}}{\gamma d}}$$

or equivalently

$$\sqrt{\frac{(1-\gamma) m \log \frac{1}{\epsilon}}{\gamma d}} + \sqrt{\frac{mk \log \frac{1}{\epsilon}}{d}} \le \frac{C}{|\mathcal{A}|}.$$

This is satisfied for $\epsilon = \exp(-C \cdot \frac{d}{k(1+m|\mathcal{A}|^2)})$ for some different constant $C$ when $\gamma = \Omega(\frac{1}{k})$, which is true by assumption. So if $\sqrt{\gamma} A_{\min} \ge \frac{1}{2}$, then the network has generalization error less than $\epsilon$ whenever $\omega$ occurs, which happens with probability at least $1 - \delta$.

Now suppose that $\sqrt{1-\gamma}\sum_{j=1}^{2m}|\langle \boldsymbol{z}_j, \boldsymbol{n}_i\rangle| \geq \frac{1}{2}$ for all $i \in \mathcal{G}_s$. Squaring both sides of the inequality and applying Cauchy-Schwarz, we obtain

$$\frac{1}{4} \leq (1-\gamma)\left(\sum_{j=1}^{2m}|\langle \boldsymbol{z}_j, \boldsymbol{n}_i\rangle|\right)^2$$

$$\leq 2m(1-\gamma)\sum_{j=1}^{2m}|\langle \boldsymbol{z}_j, \boldsymbol{n}_i\rangle|^2.$$

Summing over all $i \in \mathcal{G}_s$, we obtain

$$\frac{|\mathcal{G}_s|}{4} \leq 2m(1-\gamma)\sum_{i\in\mathcal{G}_s}\sum_{j=1}^{2m}|\langle \boldsymbol{z}_j, \boldsymbol{n}_i\rangle|^2$$

$$= 2m(1-\gamma)\sum_{j=1}^{2m}\|\boldsymbol{N}_{\mathcal{G}_s}\boldsymbol{z}_j\|^2,$$

Applying $\omega_2$ and $\omega_3$, we obtain the bound

$$n \leq Cm(1-\gamma)\sum_{j=1}^{2m}\|\boldsymbol{N}_{\mathcal{G}_s}\boldsymbol{z}_j\|^2$$

$$\leq Cm(1-\gamma)\sum_{j=1}^{2m}\|\boldsymbol{N}_{\mathcal{G}_s}\|^2\|\boldsymbol{z}_j\|^2$$

$$\leq Cm(1-\gamma)\sum_{j=1}^{2m}\|\boldsymbol{z}_j\|^2$$

$$= Cm(1-\gamma)\|\boldsymbol{Z}\|_F^2$$

$$\leq Cm(1-\gamma)\|\boldsymbol{W}\|_F^2.$$

Then applying (20),

$$n \leq Cm(1-\gamma)\|\boldsymbol{W}\|_F^2$$

$$\leq Cm(1-\gamma)|\mathcal{A}|^2\left(\frac{1}{\gamma} + \frac{k}{1-\gamma}\right)$$

$$\leq Cm|\mathcal{A}|^2\left(\frac{1}{\gamma} + k\right).$$

This implies that

$$n \leq \frac{Cm|\mathcal{A}|^2}{\gamma}$$

or

$$k \geq \frac{Cn}{m|\mathcal{A}|^2}.$$

Neither of these conditions can occur if $\gamma = \Omega\left(\frac{1}{k}\right)$ and $k = O\left(\frac{n}{|\mathcal{A}|^2 m}\right)$. Thus, in all cases, the network has generalization error less than $\exp(-C \cdot \frac{d}{k(1+m|\mathcal{A}|^2)})$ when $\omega$ occurs, which happens with probability at least $1 - \delta$. $\qquad\square$

We are also able to show the lower bound for the generalization error stated in the main text.

**Theorem 3.3.** *Under the setting given in Assumption 2, let $\delta \in (0,1)$ and suppose $\mathcal{A} = \mathcal{A}_{GD}$ where $\eta, \lambda \in \mathbb{R}_{>0}$ satisfy Assumption 1. If $n = \Omega(k)$, $d = \Omega(n)$, and $k = \Omega(\log\frac{1}{\delta} + \frac{1}{\alpha})$, then there is a fixed positive constant $C$ such that with probability at least $1 - \delta$ over $(\boldsymbol{X}, \hat{\boldsymbol{y}})$*

$$\mathbb{P}(yf(\boldsymbol{W}, \boldsymbol{x}) \leq 0 \mid \boldsymbol{X}, \hat{\boldsymbol{y}}) \geq \exp\left(-C \cdot \frac{d}{\alpha k}\right).$$

*Proof.* We proceed along the lines of Theorem 3.2. For $s \in \{1, -1\}$, let $\mathcal{B}_s$ denote the set of $i \in \mathcal{B}$ such that $\langle v, x_i \rangle = s$. Note $|\mathcal{B}| = \Omega\left(\log \frac{1}{\delta}\right)$. Under these assumptions, by Lemma D.3,

$$|\mathcal{B}_s| \geq \frac{1}{4}|\mathcal{B}| \geq Ck$$

for both $s \in \{1, -1\}$ with probability at least $1 - \frac{\delta}{3}$. We denote this event by $\omega_1$. For $s \in \{1, -1\}$, let $N_{\mathcal{B}_s} \in \mathbb{R}^{|\mathcal{B}_s| \times d}$ be the matrix whose rows are indexed by $\mathcal{B}_s$ and are given by the vectors $n_i$ for $i \in \mathcal{B}_s$. As $d = \Omega(n) = \Omega\left(k + \log \frac{1}{\delta}\right)$ and the rows of $N_{\mathcal{B}_s}$ are drawn mutually i.i.d. from $\mathcal{N}(0_d, d^{-1}(I_d - v^T))$, the following holds by Lemma 4.1. With probability at least $1 - \frac{\delta}{3}$ over the randomness of the training data, $\|N_{\mathcal{B}_s}\| \leq C$ for both $s \in \{1, -1\}$. We denote this event by $\omega_2$. By Lemma D.1, there is a constant $C$ such that

$$\|Z\|_F^2 - 2\lambda\sqrt{2m}\|Z\|_F - 2m\lambda^2 \leq \frac{C}{\alpha m}\left(\|z_{\text{lin}}\| + 2m\lambda\right)^2. \tag{21}$$

with probability at least $1 - \frac{\delta}{3}$. We denote this event by $\omega_3$. Let $\omega = \omega_1 \cap \omega_2 \cap \omega_3$. By the union bound $\mathbb{P}(\omega) \geq 1 - \delta$. We condition on $\omega$ for the remainder of this proof.

Let $s \in \{1, -1\}$ be such that $A_s = \max\{A_1, A_{-1}\}$. Since the network attains zero loss, for all $i \in \mathcal{B}_s$,

$$1 \leq \hat{y}_i f(W, x_i)$$

$$= \sum_{j=1}^{2m} (-1)^j \hat{y}_i \sigma(\langle w_j, x_i \rangle)$$

$$= \sum_{j=1}^{2m} (-1)^j y_i \sigma(\langle a_j v + z_j, -\sqrt{\gamma} y_i v + \sqrt{1-\gamma} n_i \rangle)$$

$$= \sum_{j=1}^{2m} (-1)^j y_i \sigma(-\sqrt{\gamma} a_j y_i + \sqrt{1-\gamma}\langle z_j, n_i \rangle)$$

$$\leq \sum_{j=1}^{2m} (-1)^j y_i \sigma(-\sqrt{\gamma} a_j y_i) + \sum_{j=1}^{2m} |\sqrt{1-\gamma}\langle z_j, n_i \rangle|$$

$$= \sqrt{\gamma}\sum_{j=1}^{2m} (-1)^{j+1} s\sigma(sa_j) + \sqrt{1-\gamma}\sum_{j=1}^{2m} |\langle z_j, n_i \rangle|$$

$$= -\sqrt{\gamma} A_s + \sqrt{1-\gamma}\sum_{j=1}^{2m} |\langle z_j, n_i \rangle|.$$

From which we conclude

$$\sqrt{1-\gamma}\sum_{j=1}^{2m} |\langle z_j, n_i \rangle| \geq 1 + \sqrt{\gamma} A_s \geq \sqrt{\gamma} A_s$$

for all such $i$. Squaring both sides of the inequality and applying Cauchy-Schwarz, we obtain

$$\gamma A_s \leq (1-\gamma)\left(\sum_{j=1}^{2m} |\langle z_j, n_i \rangle|\right)^2$$

$$\leq 2m(1-\gamma)\sum_{j=1}^{2m} |\langle z_j, n_i \rangle|^2.$$

Summing over all $i \in \mathcal{B}_s$, we obtain

$$|\mathcal{B}_s|\gamma A_s \leq 2m(1-\gamma)\sum_{i \in \mathcal{B}_s}\sum_{j=1}^{2m} |\langle z_j, n_i \rangle|^2$$

$$= 2m(1-\gamma)\sum_{j=1}^{2m} \|N_{\mathcal{B}_s} z_j\|^2,$$

Applying $\omega_2$ and $\omega_3$, we obtain the bound

$$k\gamma A_s \le Cm(1-\gamma) \sum_{j=1}^{2m} \|\boldsymbol{N}_{\mathcal{B}_s} \boldsymbol{z}_j\|^2$$

$$\le Cm(1-\gamma) \sum_{j=1}^{2m} \|\boldsymbol{N}_{\mathcal{B}_s}\|^2 \|\boldsymbol{z}_j\|^2$$

$$\le Cm(1-\gamma) \sum_{j=1}^{2m} \|\boldsymbol{z}_j\|^2$$

$$= Cm(1-\gamma) \|\boldsymbol{Z}\|_F^2.$$

For $k = \Omega(\frac{1}{\alpha})$, this inequality along with Assumption 1 implies that

$$C\|\boldsymbol{Z}\|_F^2 \le \|\boldsymbol{Z}\|_F^2 + 2\lambda\sqrt{2m}\|\boldsymbol{Z}\|_F + 2m\lambda^2$$

for a different constant $C$. With the last two inequalities and equation 21, we obtain the bound, for a new constant $C$.

$$k\gamma A_s \le C\frac{1-\gamma}{\alpha} \left(\|\boldsymbol{z}_{\text{lin}}\| + 2m\lambda\right)^2.$$

We then apply $k = \Omega(\frac{1}{\alpha})$ and Assumption 1 again to conclude that for some $C$,

$$\|\boldsymbol{z}_{\text{lin}}\| \ge C\sqrt{\frac{k\gamma A_s \alpha}{1-\gamma}}.$$

Note that

$$A_{\text{lin}} = \frac{A_1 + A_{-1}}{1+\alpha} \le 2A_s.$$

We then bound

$$\frac{A_{\text{lin}}}{\|\boldsymbol{z}_{\text{lin}}\|} \le C\frac{A_s}{\sqrt{\frac{k\gamma A_s \alpha}{1-\gamma}}}$$

$$\le C\sqrt{\frac{1-\gamma}{k\gamma\alpha}}$$

for some constant $C$. Now consider a test point $(\boldsymbol{x}, y)$, which satisfies

$$\boldsymbol{x} = y(\sqrt{\gamma}\boldsymbol{v} + \sqrt{1-\gamma}\boldsymbol{n}),$$

where $\boldsymbol{n} \sim \mathcal{N}(\boldsymbol{0}_d, \frac{1}{d}(\boldsymbol{I}_d - \boldsymbol{v}\boldsymbol{v}^T))$. Since the data distribution is symmetric,

$$\mathbb{P}(yf(\boldsymbol{W}, \boldsymbol{x}) \le 0) \ge \frac{1}{2}\mathbb{P}(yf(\boldsymbol{W}, \boldsymbol{x}) \le 0 \text{ or } -yf(\boldsymbol{W}, -\boldsymbol{x}) \le 0)$$

$$\ge \frac{1}{2}\mathbb{P}(yf(\boldsymbol{W}, \boldsymbol{x}) - yf(\boldsymbol{W}, -\boldsymbol{x}) \le 0).$$

We see that

$$yf(\boldsymbol{W}, \boldsymbol{x}) - yf(\boldsymbol{W}, -\boldsymbol{x}) = (1+\alpha)\left(yA_{\text{lin}}\sqrt{\gamma} + \langle \boldsymbol{z}_{\text{lin}}, \boldsymbol{n}\rangle\sqrt{1-\gamma}\right)$$

By Lemma C.1 we then can bound

$$\mathbb{P}(y\langle \boldsymbol{w}, \boldsymbol{x}\rangle \le 0 \mid \omega) \ge \frac{1}{8}\exp\left(-\frac{6d}{\pi}\frac{\gamma}{1-\gamma}\frac{A_{\text{lin}}^2}{\|\boldsymbol{z}_{\text{lin}}\|^2}\right)$$

$$\ge \exp\left(-\frac{Cd}{\alpha k}\right)$$

for a new constant $C$, provided $A_{\text{lin}}$ is positive. In the last line we can bound $\frac{d}{k}$ below as $d = \Omega(n) = \Omega(k)$. If $A_{\text{lin}}$ is negative, then the generalization error is at least $\frac{1}{4}$ which is also at least $\exp(-Cd/(\alpha k))$. $\qquad\square$

### D.3 Non-benign overfitting

In this section we show that leaky ReLU networks trained on low-signal data exhibit non-benign overfitting. As in the case of benign overfitting, we will rely on a generalization bound which depends on the signal-to-noise ratio of the network.

**Lemma D.4.** *Let $W \in \mathbb{R}^{2m \times d}$ be the first layer weight matrix of a shallow leaky ReLU network given by equation 1. Suppose $(x, y)$ is a random test point sampled under the data model given in Definition 2.1. If $W$ is such that $A_{\mathrm{lin}} \geq 0$ and*

$$\frac{A_{\mathrm{lin}}}{z_{\mathrm{lin}}} = O\left(\sqrt{\frac{1 - \gamma}{\gamma d}}\right)$$

*then*

$$\mathbb{P}(yf(W, x) < 0) \geq \frac{1}{8}.$$

*Alternatively, if $A_{\mathrm{lin}} \leq 0$ then*

$$\mathbb{P}(yf(W, x) < 0) \geq \frac{1}{4}.$$

*Proof.* By Definition 2.1 $(-x, -y)$ is identically distributed to $(x, y)$, therefore

$$\mathbb{P}(0 > yf(W, x)) = \frac{1}{2}\left(\mathbb{P}(0 > yf(W, x)) + \mathbb{P}(0 > -yf(W, -x))\right)$$

$$\geq \frac{1}{2}\mathbb{P}(0 > yf(W, x) \cup 0 > -yf(W, -x))$$

$$\geq \frac{1}{2}\mathbb{P}\left(0 > yf(W, x) - yf(W, -x)\right).$$

Next we compute

$$yf(W, x) - yf(W, -x)$$

$$= \sum_{j=1}^{2m}(-1)^j y(\sigma(\langle w_j, x \rangle) - \sigma(\langle w_j, -x \rangle))$$

$$= (1 + \alpha)\sum_{j=1}^{2m}(-1)^j y \langle w_j, x \rangle$$

$$= (1 + \alpha)\sum_{j=1}^{2m}(-1)^j \langle a_j v + z_j, \sqrt{\gamma} v + \sqrt{1 - \gamma} n \rangle$$

$$= (1 + \alpha)\sqrt{\gamma}\sum_{j=1}^{2m}(-1)^j a_j + (1 + \alpha)\sqrt{1 - \gamma}\left\langle n, \sum_{j=1}^{2m}(-1)^j z_j \right\rangle$$

$$= (1 + \alpha)\sqrt{\gamma}A_{\mathrm{lin}} + (1 + \alpha)\sqrt{1 - \gamma}\langle n, z_{\mathrm{lin}} \rangle.$$

The above two calculations imply that

$$\mathbb{P}(0 > yf(W, x)) \geq \frac{1}{2}\mathbb{P}(0 > yf(W, x) - yf(W, -x))$$

$$= \frac{1}{2}\mathbb{P}(0 > (1 + \alpha)\sqrt{\gamma}A_{\mathrm{lin}} + (1 + \alpha)\sqrt{1 - \gamma}\langle n, z_{\mathrm{lin}} \rangle)$$

$$= \frac{1}{2}\mathbb{P}\left(\langle -n, z_{\mathrm{lin}} \rangle > \sqrt{\frac{\gamma}{1 - \gamma}}A_{\mathrm{lin}}\right).$$

Suppose that $A_{\mathrm{lin}} \geq 0$. As the noise distribution is symmetric $\langle n, z_{\mathrm{lin}} \rangle \overset{d}{=} \langle -n, z_{\mathrm{lin}} \rangle$. Therefore,

$$\frac{1}{4}\mathbb{P}\left(|\langle n, z_{\mathrm{lin}} \rangle| > \sqrt{\frac{\gamma}{1 - \gamma}}A_{\mathrm{lin}}\right) = \frac{1}{4}\mathbb{P}\left(|\langle n, u \rangle| > \sqrt{\frac{\gamma}{1 - \gamma}}\frac{A_{\mathrm{lin}}}{\|z_{\mathrm{lin}}\|}\right),$$

where $\boldsymbol{u} = \frac{\boldsymbol{z}_{\text{lin}}}{\|\boldsymbol{z}_{\text{lin}}\|}$ is the unit vector pointing in the direction of $\boldsymbol{z}_{\text{lin}}$. Note by construction $\boldsymbol{u} \in \text{span}(\{\boldsymbol{v}\})^{\perp}$. If

$$\frac{A_{\text{lin}}}{\|\boldsymbol{z}_{\text{lin}}\|} = O\left(\sqrt{\frac{1-\gamma}{\gamma d}}\right),$$

then by Lemma A.2,

$$\mathbb{P}\left(|\langle \boldsymbol{n}, \boldsymbol{u}\rangle| > \sqrt{\frac{\gamma}{1-\gamma}} \frac{A_{\text{lin}}}{\|\boldsymbol{z}_{\text{lin}}\|}\right) \geq \frac{1}{2}$$

and therefore

$$\mathbb{P}(0 > yf(\boldsymbol{W}, \boldsymbol{x})) \geq \frac{1}{4}\mathbb{P}\left(|\langle \boldsymbol{n}, \boldsymbol{u}\rangle| > \sqrt{\frac{\gamma}{1-\gamma}} \frac{A_{\text{lin}}}{\|\boldsymbol{z}_{\text{lin}}\|}\right) \geq \frac{1}{8}.$$

If $A_{\text{lin}} < 0$, then again by the symmetry of the noise

$$\begin{aligned}
\mathbb{P}(0 > yf(\boldsymbol{W}, \boldsymbol{x})) &\geq \frac{1}{2}\mathbb{P}\left(\langle -\boldsymbol{n}, \boldsymbol{z}_{\text{lin}}\rangle > \sqrt{\frac{\gamma}{1-\gamma}} A_{\text{lin}}\right) \\
&\geq \frac{1}{2}\mathbb{P}\left(\langle -\boldsymbol{n}, \boldsymbol{z}_{\text{lin}}\rangle > 0\right) \\
&= \frac{1}{4}.
\end{aligned}$$

This establishes the result. $\qquad\square$

**Theorem 3.4.** *Under the setting given in Assumption 2, let $\delta \in (0,1)$ and suppose $\mathcal{A} = \mathcal{A}_{GD}$, where $\eta, \lambda \in \mathbb{R}_{>0}$ satisfy Assumption 1. If $n = \Omega(1), d = \Omega\left(n + \log\frac{1}{\delta}\right)$ and $\gamma = O\left(\frac{\alpha^3}{d}\right)$ then the following hold.*

1. *The algorithm $\mathcal{A}_{GD}$ terminates almost surely after finitely many updates. With $\boldsymbol{W} = \mathcal{A}_{GD}(\boldsymbol{X}, \hat{\boldsymbol{y}})$, $L(\boldsymbol{W}, \boldsymbol{X}, \hat{\boldsymbol{y}}) = 0$.*

2. *With probability at least $1 - \delta$ over the training data $(\boldsymbol{X}, \hat{\boldsymbol{y}})$*

$$\mathbb{P}(yf(\boldsymbol{W}, \boldsymbol{x}) < 0 \mid \boldsymbol{X}, \hat{\boldsymbol{y}}) \geq \frac{1}{8}.$$

*Proof.* If $A_{\text{lin}} < 0$, then by Lemma D.4,

$$\mathbb{P}(yf(\boldsymbol{W}, \boldsymbol{x}) < 0) \geq \frac{1}{4}.$$

So it suffices to consider the case $A_{\text{lin}} \geq 0$. Since $d = \Omega\left(n + \log\frac{1}{\delta}\right)$, by Lemma B.4, the max-margin classifier $\boldsymbol{w}^*$ satisfies

$$\|\boldsymbol{w}^*\| \leq C\sqrt{\frac{n}{1-\gamma}}$$

with probability at least $1 - \frac{\delta}{3}$ over the randomness of the input dataset. We denote this event by $\omega_1$ and condition on it for the rest of this proof. By Theorem 3.1,

$$\begin{aligned}
\|\boldsymbol{W}\| &\leq \frac{C\|\boldsymbol{w}^*\|}{\alpha\sqrt{m}} \\
&\leq \frac{C}{\alpha}\sqrt{\frac{n}{m(1-\gamma)}}.
\end{aligned}$$

By Theorem 3.1, the network perfectly fits the training data, so for all $i \in [n]$, $\hat{y}_i f(\boldsymbol{W}, \boldsymbol{x}_i) \geq 1$, and therefore

$$
\begin{aligned}
1 \leq &\ |f(\boldsymbol{W}, \boldsymbol{x}_i)| \\
= &\ \left| \sum_{j=1}^{2m} (-1)^j \sigma(\langle \boldsymbol{w}_j, \boldsymbol{x}_i \rangle) \right| \\
\leq &\ \sum_{j=1}^{2m} |\langle \boldsymbol{w}_j, \boldsymbol{x}_i \rangle| \\
= &\ \sum_{j=1}^{2m} |\langle a_j \boldsymbol{v} + \boldsymbol{z}_j, \sqrt{\gamma} y_i \boldsymbol{v} + \sqrt{1-\gamma} \boldsymbol{n}_i \rangle| \\
= &\ \sum_{j=1}^{2m} |a_j y_i \sqrt{\gamma} + \sqrt{1-\gamma} \langle \boldsymbol{z}_j, \boldsymbol{n}_i \rangle| \\
\leq &\ \sqrt{\gamma} \sum_{j=1}^{2m} |a_j| + \sqrt{1-\gamma} \sum_{j=1}^{2m} |\langle \boldsymbol{z}_j, \boldsymbol{n}_i \rangle|.
\end{aligned}
$$

This implies that either $\frac{1}{2} \leq \sqrt{\gamma} \sum_{j=1}^{2m} |a_j|$ or $\frac{1}{2} \leq \sqrt{1-\gamma} \sum_{j=1}^{2m} |\langle \boldsymbol{z}_j, \boldsymbol{n}_i \rangle|$ for all $i \in [n]$. We consider both cases separately.

Suppose that $\frac{1}{2} \leq \sqrt{\gamma} \sum_{j=1}^{2m} |a_j|$. Then squaring both sides and applying Cauchy-Schwarz, we obtain

$$
\begin{aligned}
\frac{1}{4} \leq &\ \gamma \left( \sum_{j=1}^{2m} |a_j| \right)^2 \\
\leq &\ 2m\gamma \sum_{j=1}^{2m} |a_j|^2 \\
\leq &\ 2m\gamma \sum_{j=1}^{2m} \|\boldsymbol{w}_j\|^2 \\
= &\ 2m\gamma \|\boldsymbol{W}\|_F^2 \\
\leq &\ \frac{C\gamma n}{\alpha^2 (1-\gamma)}.
\end{aligned}
$$

This cannot occur if $\gamma = O\left(\frac{\alpha^2}{n}\right)$, and in particular it cannot occur if $d = \Omega(n)$ and $\gamma = O\left(\frac{\alpha^3}{d}\right)$.

Now suppose that $\frac{1}{2} \leq \sqrt{1-\gamma} \sum_{j=1}^{2m} |\langle \boldsymbol{z}_j, \boldsymbol{n}_i \rangle|$ for all $i \in [n]$. Squaring both sides and applying Cauchy-Schwarz, we obtain

$$
\begin{aligned}
\frac{1}{4} \leq &\ (1-\gamma) \left( \sum_{j=1}^{2m} \|\langle \boldsymbol{z}_j, \boldsymbol{n}_i \rangle\| \right)^2 \\
\leq &\ 2m(1-\gamma) \sum_{j=1}^{2m} \|\langle \boldsymbol{z}_j, \boldsymbol{n}_i \rangle\|^2.
\end{aligned}
$$

Summing over all $i \in [n]$, we obtain

$$\frac{n}{4} \leq 2m(1-\gamma) \sum_{i=1}^{n} \sum_{j=1}^{2m} \|\langle \boldsymbol{z}_j, \boldsymbol{n}_i \rangle\|^2$$

$$= 2m(1-\gamma) \sum_{j=1}^{2m} \|\boldsymbol{N}\boldsymbol{z}_j\|^2$$

$$\leq 2m(1-\gamma)\|\boldsymbol{N}\|^2 \sum_{j=1}^{2m} \|\boldsymbol{z}_j\|^2$$

$$= 2m(1-\gamma)\|\boldsymbol{N}\|^2 \|\boldsymbol{Z}\|_F^2.$$

Recall that $d = \Omega\left(n + \log \frac{1}{\delta}\right)$, and that the rows of $\boldsymbol{N}$ are i.i.d. with distribution $\mathcal{N}(\boldsymbol{0}_d, d^{-1}(\boldsymbol{I}_d - \boldsymbol{v}\boldsymbol{v}^T))$. So by Lemma 4.1, with probability at least $1 - \frac{\delta}{3}$ over the randomness of the dataset, $\|\boldsymbol{N}\| \leq C$. We denote this event by $\omega_2$ and condition on it for the rest of this proof. So

$$\|\boldsymbol{Z}\|_F^2 \geq \frac{Cn}{m(1-\gamma)}. \tag{22}$$

Let $\lambda = \frac{\sqrt{\alpha}}{m}$. By Assumption 1, $\|\boldsymbol{w}_j^{(0)}\| \leq \lambda$ for all $j \in [2m]$. So by Lemma D.1,

$$\|\boldsymbol{Z}\|_F^2 - 2\lambda\sqrt{2m}\|\boldsymbol{Z}\|_F - 2m\lambda^2 \leq \frac{C}{\alpha m}(\|\boldsymbol{z}_{\text{lin}}\| + 2m\lambda)^2. \tag{23}$$

By (22),

$$\|\boldsymbol{Z}\|_F \geq \frac{C\sqrt{n}}{\sqrt{m(1-\gamma)}}$$

$$\geq \frac{C\sqrt{n}}{\sqrt{m}}$$

$$\geq C\lambda\sqrt{nm}$$

$$\geq 8\lambda\sqrt{2m},$$

where the last line holds if $n = \Omega(1)$. Then by (23),

$$\frac{1}{2}\|\boldsymbol{Z}\|_F^2 = \|\boldsymbol{Z}\|_F^2 - \frac{1}{4}\|\boldsymbol{Z}\|_F^2 - \frac{1}{4}\|\boldsymbol{Z}\|_F^2$$

$$\leq \|\boldsymbol{Z}\|_F^2 - 2\lambda\sqrt{2m}\|\boldsymbol{Z}\|_F - 2m\lambda^2$$

$$\leq \frac{C}{\alpha m}(\|\boldsymbol{z}_{\text{lin}}\| + 2m\lambda)^2$$

$$= \frac{C}{\alpha m}(\|\boldsymbol{z}_{\text{lin}}\| + 2\sqrt{\alpha})^2.$$

Taking the square root of both sides and recalling that $\alpha \in (0, 1)$ is a constant, we obtain

$$\|\boldsymbol{Z}\|_F \leq \frac{C\|\boldsymbol{z}_{\text{lin}}\|}{\sqrt{\alpha m}} + \frac{C}{\sqrt{m}}.$$

This implies that either $\|\boldsymbol{Z}\|_F \leq \frac{2C}{\sqrt{m}}$ or $\|\boldsymbol{Z}\|_F \leq \frac{2C\|\boldsymbol{z}_{\text{lin}}\|}{\sqrt{\alpha m}}$. The case $\|\boldsymbol{Z}\|_F \leq \frac{2C}{\sqrt{m}}$ cannot happen, since by (22),

$$\|\boldsymbol{Z}\|_F \geq \frac{C'\sqrt{n}}{\sqrt{m(1-\gamma)}}$$

$$\geq \frac{C'\sqrt{n}}{\sqrt{m}}$$

$$\geq \frac{2C}{\sqrt{m}}$$

when $n = \Omega(1)$. So we have $\|\boldsymbol{Z}\|_F \leq \frac{2C\|\boldsymbol{z}_{\text{lin}}\|}{\sqrt{\alpha m}}$. Again applying (22), we obtain

$$\|\boldsymbol{z}_{\text{lin}}\| \geq C\sqrt{\alpha m}\|\boldsymbol{Z}\|_F$$
$$\geq \frac{C\sqrt{\alpha n}}{\sqrt{1-\gamma}}.$$

So

$$\frac{A_{\text{lin}}}{\|\boldsymbol{z}_{\text{lin}}\|} \leq C\frac{A_{\text{lin}}\sqrt{1-\gamma}}{\sqrt{\alpha n}}$$
$$= \frac{C\sqrt{1-\gamma}}{\sqrt{\alpha n}}\sum_{j=1}^{2m}(-1)^j a_j$$
$$\leq \frac{C\sqrt{1-\gamma}}{\sqrt{\alpha n}}\sqrt{2m}\left(\sum_{j=1}^{2m}|a_j|^2\right)^{1/2}$$
$$\leq \frac{C\sqrt{m(1-\gamma)}}{\sqrt{\alpha n}}\left(\sum_{j=1}^{2m}\|\boldsymbol{w}_j\|^2\right)^{1/2}$$
$$= \frac{C\|\boldsymbol{W}\|_F\sqrt{m(1-\gamma)}}{\sqrt{\alpha n}}$$
$$\leq \frac{C}{\alpha^{3/2}}.$$

Here we used that $A_{\text{lin}} \geq 0$ and applied Cauchy-Schwarz in the third line. Then by Lemma D.4, if

$$\frac{C}{\alpha^{3/2}} \leq O\left(\sqrt{\frac{1-\gamma}{\gamma d}}\right),$$

then

$$\mathbb{P}(yf(\boldsymbol{W},\boldsymbol{x}) < 0) \geq \frac{1}{8}.$$

This occurs if $\gamma = O\left(\frac{\alpha^3}{d}\right)$. Hence, in all cases, we have shown that with the appropriate scaling, the generalization error is at least $\frac{1}{8}$ when both $\omega_1$ and $\omega_2$ occur. This happens with probability at least $1 - \delta$. $\qquad\square$

## Appendix E    Formalizing benign overfitting as a high dimensional phenomenon

To formalize benign overfitting as a high dimensional phenomenon we first introduce the notion of a *regime*. Informally, a regime is a subset of the hyperparameters $\Omega \in \mathbb{N}^4$ which describes accepted combinations of the input data dimension $d$, the number of points in the training sample $n$, the number of corrupt points $k$ and the number of trainable model parameters $p$.

**Definition E.1.** *A regime is a subset $\Omega \subset \mathbb{N}^4$ which satisfies the following properties.*

1. *For any tuple $(d, n, k, p) \in \Omega$ the number of corrupt points is at most the total number of points, $k \leq n$.*

2. *There is no upper bound on the number of points,*

$$\sup_{(d,n,k,p)\in\Omega} n = \infty.$$

*A non-trivial regime is a regime which satisfies the following additional condition.*

3. *Define the set of increasing sequences of $\Omega$ as $\Omega^* = \{(n_l, d_l, k_l, p_l)_{l \in \mathbb{N}} \subset \Omega \ s.t. \ \lim_{l \to \infty} n_l = \infty\}$. For any $(n_l, d_l, k_l, p_l)_{l \in \mathbb{N}} \in \Omega^*$ it holds that*

$$\liminf_{l \to \infty} \frac{k_l}{n_l} > 0.$$

Intuitively, a regime defines how the four hyperparameters $(d, n, k, p)$ can grow in relation to one another as $n$ goes to infinity. A non-trivial regime is one in which the fraction of corrupt points in the training sample is non-vanishing. In order to make a formal definition of benign overfitting as high dimensional phenomenon we introduce the following additional concepts.

- A *learning algorithm* $\mathcal{A} = (\mathcal{A}_{d,n,p})_{(d,n,p) \in \mathbb{N}^3}$ is a triple indexed sequence of measurable functions $\boldsymbol{A}_{d,n,p} : \mathbb{R}^{n \times d} \times \mathbb{R}^n \to \mathbb{R}^p$.

- An *architecture* $\mathcal{M} = (f_{d,p})_{d,p \in \mathbb{N}^2}$ is a double indexed sequence of measurable functions $f_{d,p} : \mathbb{R}^d \times \mathbb{R}^p \to \mathbb{R}$.

- A *data model* $\mathcal{D} = (D_{d,n,k})_{(d,n,k) \in \mathbb{N}^3}$ is a triple indexed sequence of Borel probability measures $D_{d,n,k}$ defined over $\mathbb{R}^{n \times d} \times \{\pm 1\} \times \mathbb{R}^d \times \{\pm 1\}$.

With these notions in place we are ready to provide a definition of benign overfitting in high dimensions.

**Definition E.2.** *Let $(\epsilon, \delta) \in (0,1]^2$, $\mathcal{A}$ be a learning algorithm, $\mathcal{M}$ an architecture, $\mathcal{D}$ a data model and $\Omega$ a regime. If for every increasing sequence $(d_l, n_l, k_l, p_l)_{l \in \mathbb{N}} \in \Omega^*$ there exists an $L \in \mathbb{N}$ such that for all $l \geq L$ with probability at least $1 - \delta$ over $(\boldsymbol{X}, \hat{\boldsymbol{y}})$, where $(\boldsymbol{X}, \hat{\boldsymbol{y}}, \boldsymbol{x}, y) \sim D_{d_l, n_l, k_l}$, it holds that*

1. *$y_i f(\mathcal{A}_{d_l, n_l, p_l}(\boldsymbol{X}, \hat{\boldsymbol{y}}), \boldsymbol{x}_i) > 0 \ \ \forall i \in [n_l]$,*

2. *$\mathbb{P}(yf(\mathcal{A}_{d_l, n_l, p_l}(\boldsymbol{X}, \hat{\boldsymbol{y}}), \boldsymbol{x}) \leq 0) \leq \inf_{\boldsymbol{W} \in \mathbb{R}^{n_l \times d_l}} \mathbb{P}(yf(\boldsymbol{W}, \boldsymbol{x}) \leq 0) + \epsilon$,*

*then the quadruplet $(\mathcal{A}, \mathcal{M}, \mathcal{D}, \Omega)$ $(\epsilon, \delta)$-benignly overfits. If $(\mathcal{A}, \mathcal{M}, \mathcal{D}, \Omega)$ $(\epsilon, \delta)$-benignly overfits for any $(\epsilon, \delta) \in (0,1]^2$ then we say $(\mathcal{A}, \mathcal{M}, \mathcal{D}, \Omega)$ benignly overfits.*

Analogously, we define non-benign overfitting as follows.

**Definition E.3.** *Let $(\epsilon, \delta) \in (0,1]^2$, $\mathcal{A}$ be a learning algorithm, $\mathcal{M}$ an architecture, $\mathcal{D}$ a data model and $\Omega$ a regime. If for every increasing sequence $(d_l, n_l, k_l, p_l)_{l \in \mathbb{N}} \in \Omega^*$ there exists an $L \in \mathbb{N}$ such that for all $l \geq L$ with probability at least $1 - \delta$ over $(\boldsymbol{X}, \hat{\boldsymbol{y}})$, where $(\boldsymbol{X}, \hat{\boldsymbol{y}}, \boldsymbol{x}, y) \sim D_{d_l, n_l, k_l}$, it holds that*

1. *$y_i f(\mathcal{A}_{d_l, n_l, p_l}(\boldsymbol{X}, \hat{\boldsymbol{y}}), \boldsymbol{x}_i) > 0 \ \ \forall i \in [n_l]$,*

2. *$\mathbb{P}(yf(\mathcal{A}_{d_l, n_l, p_l}(\boldsymbol{X}, \hat{\boldsymbol{y}}), \boldsymbol{x}) \leq 0) \geq \inf_{\boldsymbol{W} \in \mathbb{R}^{n_l \times d_l}} \mathbb{P}(yf(\boldsymbol{W}, \boldsymbol{x}) \leq 0) + \epsilon$,*

*then the quadruplet $(\mathcal{A}, \mathcal{M}, \mathcal{D}, \Omega)$ $(\epsilon, \delta)$-non-benignly overfits. If $(\mathcal{A}, \mathcal{M}, \mathcal{D}, \Omega)$ $(\epsilon, \delta)$-non-benignly overfits for any $(\epsilon, \delta) \in (0,1]^2$ then we say $(\mathcal{A}, \mathcal{M}, \mathcal{D}, \Omega)$ non-benignly overfits.*

One of the key contributions of this paper is proving $(\epsilon, \delta)$-benign and non-benign overfitting when the architecture is a two-layer leaky ReLU network (equation 1), the learning algorithm returns the inner layer weights of the network by minimizing the hinge loss over the training data using gradient descent (Definition 2.2), and the regime satisfies the conditions $d = \Omega(n \log 1/\epsilon)$, $n = \Omega(1/\delta)$, $k = O(n)$ and $p = 2dm$ for some network width $2m$, $m \in \mathbb{N}$.

## Appendix F   Experiments

To further support our theory, we train shallow neural networks on the data model described in Definition 2.1 and record the numerical results. Scripts to reproduce these experiments can be found at `https://github.com/kedar2/benign_overfitting`. These experiments were run on the CPU of a MacBook Pro M2 with 8GB of RAM. For our first experiment, we investigate the effect of the ratio $\frac{d}{n}$ on the generalization error of the network. Recall that by Theorem 3.2, the generalization

error is bouned above by $\exp(-Cn)$ when $\frac{d}{n} = \Omega(1)$. In other words, if $\frac{d}{n}$ is larger than a critical threshold, then the generalizatione error decays quickly to 0 as $n$ increases. We empirically confirm this prediction in Figure 1, where we train several networks while varying $\frac{d}{n}$ and $n$, and estimate the generalization error for each configuration by averaging over 20 trials. Within each trial, we trained the inner layer of the network with gradient descent using the hinge loss until the training loss reached 0. For $\frac{d}{n}$ greater than around 7, the generalization error rapidly decays to 0 as $n \to \infty$.

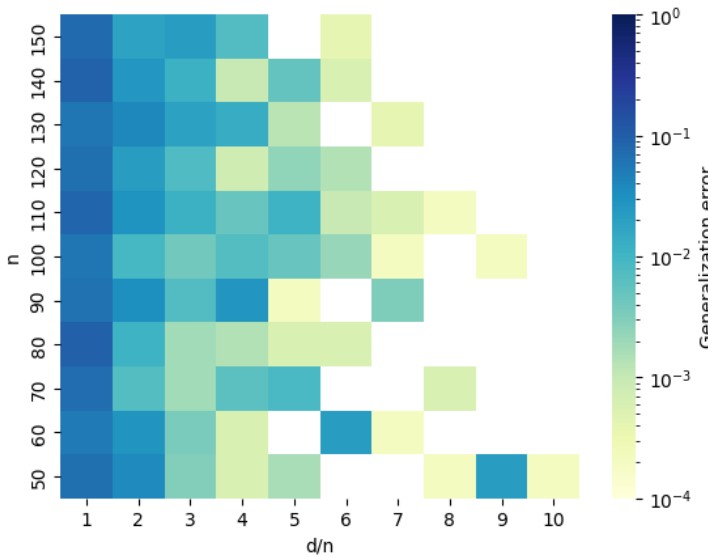

Figure 1: Generalization error of a two-layer leaky ReLU network trained to 0 hinge loss varying $n$ and $d$. Parameter settings: $\alpha = 0.1$, $\gamma = 5/n$, $m = 64$, $k = 0.1n$, number of trials $= 5$, size of validation sample $= 1000$.

Next, we train a two-layer network varying $n$ and $\gamma$ (Figure 2), holding constant the ratio $\frac{d}{n}$. Since $\gamma$ controls the signal-to-noise ratio of the data, the generalization error of the learned network decreases as $\gamma$ increases. For each value of $n$, the generalization error falls off steeply as $\gamma$ reaches a certain threshold. This threshold decreases as $n$ increases, indicating that the network has higher noise tolerance as $n$ increases. This is in agreement with our theoretical results where we found that benign overfitting occurs at the threshold $\gamma = \Omega\left(\frac{1}{k}\right)$ (which is in this case $\Omega\left(\frac{1}{n}\right)$). We also see that the generalization error for large values of $\gamma$ is similar across different values of $n$. This effect is also predicted by Corollary 3.2.1, since we scale both $d$ and $k$ proportionally to $n$.

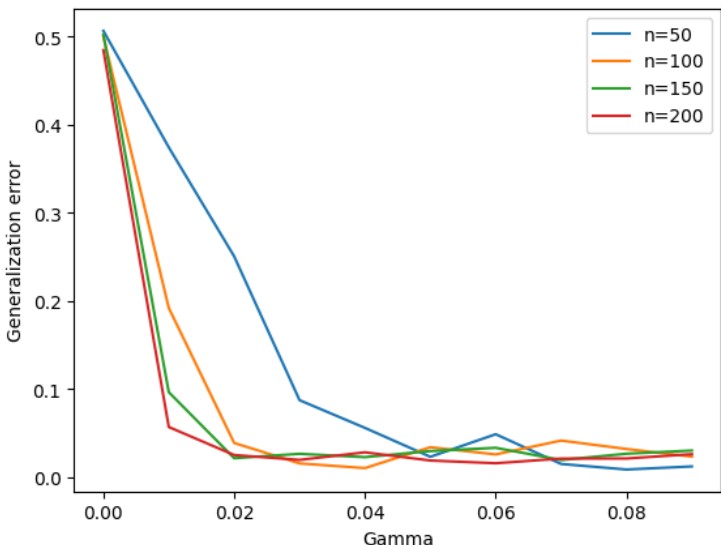

Figure 2: Generalization error of a two-layer leaky ReLU network trained to $0$ hinge loss varying $\gamma$ and $n$. Parameter settings: $\alpha = 0.1$, $d = 2n$, $m = 64$, $k = 0.1n$, number of trials $= 10$, size of validation sample $= 1000$.

