# OpenReview forum: "Benign overfitting in leaky ReLU networks with moderate input dimension"
_NeurIPS.cc/2024/Conference — NeurIPS 2024 spotlight_

### Official Review · Reviewer_xTBG · 2024-07-11

**Soundness:** 3
**Presentation:** 3
**Contribution:** 3
**Rating:** 5
**Confidence:** 4

**Summary:**

This paper stuides the benign overiftting of two-layer neural networks (only training the first layer) with leaky ReLU for binary classification, which relaxes the dimension condition on the dimension from $d = \Omega(n^2 log n)$ to $d = \Omega(n)$.

The considered problem setting is
- data generation process: the label is generated by Gaussian data corrupted by flipped noise under a linear function after sign. The parameter $\gamma$ controls the component of signal and noise. The more $\gamma$, the more signal.
- the used loss is hinge loss, and the involved with optimization algorithm is sub-gradient descent.

The obtained results include that
- implicit bias: the obtained solution (neural network parameter) will converge to a max-margin linear classifier.
- condition of benign overfitting: $d = \Omega(n)$, signal strength $\gamma = \Omega(1/k)$
- non-benign overfitting: $d = \Omega(n)$, $\gamma = O(1/d)$

**Strengths:**

- relax the dimension dependence from $n^2$ to $n$
- provide the condition for benign overfitting (or not)
- demonstrate the implicit bias: converge to a max-margin linear classifier

**Weaknesses:**

- this paper requires linear separable data, more strictly speaking, the optimal Bayes classifier is linearly separable, that means linear classifier is sufficient to learn this problem. I don't see the strong motivation of using two-layer neural networks, though this is a common issue in the benign overfitting community.

- One major issue is, no comparison with [1]. I think this comparison should be included in terms of problem setting, proof techniques, the obtained findings.

- About the main result, intuitively, classification rate is proportional to $k$, the number of corrupt points. More discussion on $\gamma$ and $k$ is needed.

- Regarding theorem 3.2, it's unclear to me in several points:
i) the sample complexity is $\Omega(n^2)$ in Theorem 3.1. How does this contribute to $|A|$?
iI) a larger $m$ leads to a larger missclassification probability. Intuitively, a larger size of neural network is better for performance.

- In Corollary 3.2.1, I didn't see $\delta$ in the main result.

- Lemma B.1 can be directly obtained from the high dimensional probability book, Chapter 5.


---
[1] George etal. Training shallow ReLU networks on noisy data using hinge loss: when do we overfit and is it benign? NeurIPS 2023.

**Questions:**

- The key idea is to bound the weight norm and its alignment with any linear separator of the data during each update by the distribution of sigular values of the noise. In this case, the obtained results heavily depend on the data-generation process. It would be possible to consider general data generation process? Besides, $d = \Omega(n)$ is still large, what is the limit for this? Maybe constant order is sufficient?

- In the proof, line 706, in the last inequality, regarding the summation $\sum_{i,l} \in F(t)$, there is one term missing about $\langle w_j^{(t)}, x_j \rangle$?

- There are some typos: line 707: $\sigma(s)$ -> $\sigma(z)$

---

> ### Author Rebuttal · Authors · 2024-08-07
>
> We thank the reviewer for the overall positive feedback on our work. We are confident we have addressed each of the issues raised and hope the reviewer might consider raising their score in light of our responses.
> >Linearly separable data
>
> Indeed, a linear classifier is sufficient to learn this problem: we also demonstrate that the max-margin linear classifier benignly overfits in our data setting. However, shallow neural networks are substantially more complex models and can express a rich class of functions. Without explicit regularization, previous conventional machine learning wisdom suggests neural networks will not learn a linear classifier. Results on benign overfitting in the linearly separable setting, including ours, demonstrate that the implicit bias of neural networks trained with gradient descent is favorable to learning linear problems.
> >No comparison with [1]
>
> We thank the reviewer for bringing this to our attention and will more thoroughly compare our work with [1] in the future revisions. Here are the comparisons with [1]:
>
> - The settings are very similar with regards to the data assumptions, network model, and training. The most significant differences is that [1] uses ReLU activations and requires $d = \Omega(n^2\log n)$ while we use leaky ReLU and $d = \Omega(n)$.
>
> - The proof techniques for the two papers are very different. In [1], the authors analyze the activation patterns of the neurons during various phases of training. We do not consider the activation patterns and instead bound how much the neural network can diverge from a linear model. These different techniques are appropriate in different settings: for a ReLU network it is possible that the dynamics of different neurons are driven by different subsets of the data, but with leaky ReLU this effect is weaker and the network is closer to a linear model.
>
> - Both papers prove the existence of multiple distinct "overfitting" regimes. However, [1] also shows a regime where the network fails to overfit. This regime does not occur for the leaky ReLU network model we use: Theorem 3.1 shows that networks always achieves zero loss in our setting.
>
> >More discussion on $\gamma$ and $k$ is needed
>
> We highlight that in both Corollary 3.2.1 and Theorem 3.3 the classification error increases with $k$. The classification error bound uses the norm of the network weights, which by Theorem 3.1 is proportional to the norm of the max margin classifier. We bound this twice. In Lemma B.3 we get a dependence on $k$ by constructing an "ideal" linear classifier: one that uses the true signal vector for clean points and the noise only as minimally needed to fit the corrupted points. If $\gamma$ is small, then the classifier constructed in Lemma B.4 (which uses only noise) has smaller norm. In fact, if $\gamma$ is too small, this norm is so much smaller that the network can never learn the ``ideal'' classifier, which leads to Theorem 3.4 (non-benign overfitting).
> >Sample complexity is $\Omega(n^2)$ in Theorem 3.1
>
> The number of iterations $T$ does not directly contribute to $|\mathcal{A}|$. In the proof of Theorem 3.1, we bound $|\mathcal{A}|$ by bounding the number of non-zero data point updates. The bound for the number of updates is also a (pessimistic) bound for $T$, which we do not use but record for completeness.
> >Larger $m$ leads to a larger misclassification probability
>
> The misclassification probability can increase if $m|\mathcal{A}|^2$ increases. But in Theorem 3.1, $|\mathcal{A}_{GD}| \sim m^{-1/2}$. Thus our results specifically for neural networks (Corollary 3.2.1, Theorems 3.3 and 3.4) are independent of $m$.
> >In Corollary 3.2.1, I didn't see $\delta$ in the main result.
>
> The parameter $\delta$ refers to the "failure probability" of the data generation process. It appears in Theorem 3.2 in lines 228--230.
> >Lemma B.1 can obtained from the high dimensional probability book
>
> We thank the reviewer for drawing our attention to Theorem 4.6.1 in High Dimensional Probability by Vershynin. We emphasize that Lemma B.1 summarizes two results by Vershynin and co-authors that together are equivalent to 4.6.1. We have clearly cited both works and do not claim this lemma to be a novel or significant part of our contribution.
> >It would be possible to consider general data generation process?
>
> Proving results for arbitrary data distributions might be intractable as benign overfitting does not always occur even experimentally. However, our results for benign overfitting (Theorem 3.2) can be proven in a slightly more general setting. If all coordinates of noise are i.i.d. sub-Gaussian, the same result will hold with an analogous proof. We chose to work with Gaussian noise to prove error lower bounds (Theorems 3.3 and 3.4). We emphasize that our data assumptions are standard for works in this space.
> >$d = \Omega(n)$ is still large, what is the limit for this? Maybe constant order is sufficient?
>
> We believe that the regime $d = O(n)$ will lead to different results than the ones observed in our paper, is challenging, and will involve different techniques to study. In particular, if $md \leq n$, then the network is underparameterized and will likely not perfectly fit the data. When $d = 1$ and the network is sufficiently wide, there are partial results in the literature which suggest that the network will perfectly fit the data, but the overfitting is not benign [2]. To our knowledge, the general case $d = O(n)$ remains open.
> >In the proof, line 706...
>
> Thank you for highlighting this. There was a missing factor of $\dot{\sigma}(\langle w\_j^{(t)}, x\_l\rangle)$. This does not affect the correctness and we will correct this in future revisions.
> >Typos
>
> Thank you for drawing this to our attention. We will make another thorough proofreading to fix typos in revised versions.
>
> [2] Guy Kornowski, Gilad Yehudai, and Ohad Shamir. From tempered to benign overfitting in ReLU neural networks. *Advances in Neural Information Processing Systems*, 2023.

---

> ### Comment · Reviewer_xTBG · 2024-08-08
>
> thanks for the authors' rebuttal.
>
> It addressed some of my concerns but there are still some issues that prevent me to give higher score for very positive support. For example, the linear separable data assumption and the specific data generation process.
>
> Besides, the comparison with [1] include different techniques but the setting is quite close.
>
> Accordingly, I kept my score unchanged.

---

> > ### Author Response · Authors · 2024-08-13
> >
> > Thank you for the thoughtful discussion. We would like to emphasize two points in response to your remaining critiques.
> >
> > - First, in regard to the linear separability and data assumptions you rightly highlight that these are clear areas for improvement. However, we would highlight i) these assumptions are typical of the prior works, and ii) in the prior works a different but still key weakness is the unrealistic assumption on the dimension of the input data. Our emphasis was on solving ii) not i) and given the technical challenges involved it seemed reasonable to keep other aspects of the problem as simple as possible.
> >
> > - Second, compared with [1], although many aspects of the setting are similar we again our improvements on the input dimension. In addition, the change from ReLU to leaky ReLU not only necessitates surprisingly different techniques but also results in different overfitting behaviour. Indeed, in [1] it is identified that there are three possible different outcomes depending on $\gamma$: harmful overfitting, benign overfitting and no overfitting. In comparison, with leaky ReLU there are only two possible outcomes depending on $\gamma$ as no overfitting is not possible. As a result, this work and [1] differ not only in terms of the setting, namely input dimension regime and the activation function used, but also have distinct and complementary takeaways.

---

### Official Review · Reviewer_7S96 · 2024-07-12

**Soundness:** 3
**Presentation:** 2
**Contribution:** 3
**Rating:** 5
**Confidence:** 3

**Summary:**

The paper studies benign overfitting in leaky ReLU networks trained with hinge loss on a binary classification task. The paper gives the conditions on the signal-to-noise ratio under which benign or harmful overfitting occurs for leaky ReLU networks. Unlike the previous related works, this paper does not require the training data to be nearly orthogonal and reduces the input dimension required from $\Omega(n^2\log n)$ to $\Omega(n)$.

**Strengths:**

1. The paper demonstrates that leaky ReLU networks trained on hinge loss with gradient descent satisfy an approximate margin maximization property.
2. Prior works usually supposed nearly orthogonal data setting and dimension $d=\Omega(n^2\log n)$. However, this paper needs linearly separable data and dimension $d=\Omega(n)$, which weakens the requirement.

**Weaknesses:**

1. The paper may lack some experiments. The results are fully theoretical. If there are some empirical experiments, the conclusion will be more convincing.
2. There are few explanations for theorems and assumptions. Some additional explanations may let readers understand the results better.

**Questions:**

1. Since hinge loss may not be so commonly used in the analysis of neural networks, maybe you can also try on other loss functions like cross-entropy loss or logistic loss.
2. Since the proof techniques and ideas are closely related to [1], what is the main difference between the two papers?

[1] Alon Brutzkus, Amir Globerson, Eran Malach, and Shai Shalev-Shwartz. SGD learns over-parameterized networks that provably generalize on linearly separable data. In International Conference on Learning Representations, 2018.

**Limitations:**

The authors have shown the limitations as the future directions in the paper.

---

> ### Author Rebuttal · Authors · 2024-08-07
>
> We thank the reviewer for reading our paper and for raising their questions and comments. We are confident that we can clarify the issues raised and hope in light of our responses below that the reviewer will consider raising their score.
> > The paper may lack some experiments. The results are fully theoretical. If there are some empirical experiments, the conclusion will be more convincing.
>
> We appreciate the suggestion of including experiments to better explain our results.  We will update the paper to include experiments in future revisions.  We've attached a file showcasing a couple of experiments along with explanations in the main rebuttal.
> > There are few explanations for theorems and assumptions. Some additional explanations may let readers understand the results better.
>
> We are happy to provide more commentary in future revisions. To clarify here, in regards to the assumptions, Assumption 1 upper bounds both the step size and initialization scale. Both of these bounds are required for the proof of convergence to a global minimizer: we remark that although they are no more egregious than comparative theory works in this space, they are typically smaller than what is used in practice. Assumption 2 clarifies the model we consider and the assumptions on the data: in particular we consider data drawn from a mixture of Gaussians with a relatively small number of label corruptions. This is very standard in the literature. In terms of our results, we give an overview in Section 1.1 outlining the role of the different Theorems and Lemmas. In terms of explaining our proof techniques, we chose in Section 4 to present the key ideas using the simple case study of a linear model. In particular we highlight four key steps: first derive generalization bounds using the SNR of a classifier in terms of the size of its component in the noise versus signal subspace, second upper bound the max margin classifier, third and fourth use the bound on the max margin classifier as well as the zero loss property to to upper and lower bound the SNR.
> > Since hinge loss may not be so commonly used in the analysis of neural networks, maybe you can also try on other loss functions like cross-entropy loss or logistic loss.
>
> We agree that different loss functions would be an interesting generalization of our results. We chose the hinge loss because it exhibits similar properties to other classification losses (such as de-emphasizing points which have already been correctly classified) while having favorable properties for theoretical analysis (such as convergence in finitely many iterations). We expect that our results could admit generalizations to the cross-entropy or logistic losses with an appropriate analogue of the convergence analysis in Theorem 3.1. Fundamentally, our generalization bounds depend on gradient descent converging to a solution which is approximately margin maximizing. For any other loss function satisfying this property, we immediately obtain a benign overfitting result by applying Theorem 3.2.
>
> > Since the proof techniques and ideas are closely related to [1], what is the main difference between the two papers?
>
> The main result of [1] is that  a leaky ReLU neural network trained with the hinge loss on linearly separable data will converge to a global minimizer. Moreover, the weights of this solution have norm bounded above by the norm of the max-margin linear classifier of the data. While this idea and proof technique are important to our paper, our main contributions are in describing how this margin maxmimization results in benign overfitting. That is, we are interested in the generalization of leaky ReLU networks on noisy data, and we use a similar convergence analysis as an intermediate result to this goal. While [1] does establish a generalization bound in Theorem 4, the bound assumes that *population dataset* is linearly separable rather than just the training dataset. Hence, it cannot be applied when the training dataset has label-flipping noise, which is the setting that we are interested in for benign overfitting. The bound also gets worse as the learning rate $\eta$ goes to 0, while our generalization bounds do not have any dependence on $\eta$. In order to make these adaptations, we perform an analysis which uses properties of our specific data model. In the process, we obtain a more detailed understanding of how generalization depends on the number of data points, the dimensionality of the data, and the amount of noise in the data.

---

### Official Review · Reviewer_8RfS · 2024-07-12

**Soundness:** 4
**Presentation:** 4
**Contribution:** 4
**Rating:** 8
**Confidence:** 4

**Summary:**

This paper studies the benign overfitting of two-layer leaky ReLU network for binary classification with only mild overparameterization under a simple Gaussian mixture model assumption. First, the paper proves that for sufficiently small initialization, gradient descent with hinge loss converges in a polynomial number of iterations to an approximate max-margin solution. Then, the paper establishes that any approximate margin maximizer achieves benign overfitting, or low test error, in an (essentially tight parameter) regime where $d = \Omega(n)$. These results are matched by lower bounds in certain parameter regimes (low SNR and high label noise).

**Strengths:**

1. This paper is well-written, clear, and technically interesting. I especially appreciated the technical overview, where the main proof ideas are explained.
2. Previous works required $d = \Omega(n^2 \log n)$ to obtain benign overfitting due to using near-orthogonality of the input data, but this work only requires $d = \Omega(n)$, which is tight (for the overparameterized regime). Hence extending benign overfitting to this setting requires using subgaussian bounds on the extreme singular values of Gaussian matrices. I do want to note that this idea has also been used to prove benign overfitting for binary and multiclass classification $d = \omega(n)$ (see e.g. [1, 2])
3. Many prior works studied linear models, whereas this paper studies ReLU activation (although with only one trainable linear layer).
4. It is of particular note that the proof techniques do not rely on exact margin maximization, as the results then apply to parameters found by GD in polynomial time (as opposed to the infinite time limit). Many (though not all) previous works have explicitly studied the limiting parameters (e.g. min $\ell_2$-norm interpolation), which is not reached in practice.
5. As I described in the summary, the upper bounds have matching lower bounds in some parameter regimes. Theorem 3.3 is a matching lower bound for the misclassification probability in the regime where the label noise rate $k/n = \Omega(1)$, which is fairly realistic.
[1] Wang, Ke, and Christos Thrampoulidis. "Binary classification of gaussian mixtures: Abundance of support vectors, benign overfitting, and regularization." SIAM Journal on Mathematics of Data Science 4.1 (2022): 260-284.
[2] Wu, David, and Anant Sahai. "Precise asymptotic generalization for multiclass classification with overparameterized linear models." Advances in Neural Information Processing Systems 36 (2024).

**Weaknesses:**

No major weaknesses to report.

**Questions:**

1. What do the authors expect to change for multiclass classification with a more complicated mixture model?
2. Line 278, there is a typo; it should be $X \sim N(\sqrt{\gamma}a_v, \frac{1-\gamma}{d}\| z\|^2)$.
3. Line 325, “of” is missing in “The proof Theorem 3.3,...”

**Limitations:**

Yes

---

> ### Author Rebuttal · Authors · 2024-08-07
>
> We are glad the reviewer appreciates the contribution of our work and thank them for highlighting the two related works on linear models [1,2]. We are happy to cite these and discuss them in future revisions. We also thank the reviewer for spotting the two typos listed, we will do a further thorough proofread before uploading a revised version.
>
> In regard to the question
> >What do the authors expect to change for multiclass classification with a more complicated mixture model?'
>
> A multiclass classification setting is an interesting future direction to look into.  It would involve looking at a different model / loss but the general form of argument, i.e., bounding the singular values of the noise matrix and comparing to the one hot class max-margin classifiers, seems possible to generalize to this new setting.

---

> > ### Comment · Reviewer_8RfS · 2024-08-07
> >
> > Thanks for the response, indeed it would be interesting to see how the techniques would transfer to the multiclass setting.

---

### Official Review · Reviewer_1LbK · 2024-07-12

**Soundness:** 3
**Presentation:** 3
**Contribution:** 3
**Rating:** 4
**Confidence:** 3

**Summary:**

This paper studies benign overfitting in a two-layer leaky ReLU network trained with hinge loss for a binary classification task. This paper proves that in a finite iteration, the leaky ReLU network can reach zero training loss through gradient descent, and the network weight matrix after convergence will approximate the max-margin. And this paper provides the conditions for benign overfitting and non-benign overfitting of the leaky ReLU network.

**Strengths:**

This paper improves the previous work's requirements for almost orthogonal training data and input dimension $d= \Omega(n² \log n)$, and only requires $d = \Omega(n)$.

**Weaknesses:**

This paper mainly studies a specific shallow leaky ReLU network. The applicability of the results is limited to deeper neural networks or other complex models such as CNN and transformer.

The results rely on a specific linearly separable data distribution assumption.

**Questions:**

The dependence of Theorem 3.2 and Theorem 3.3 on n, d, k does not cover all cases. For other cases, what are the characteristics of overfitting?

**Limitations:**

See Weaknesses

---

> ### Author Rebuttal · Authors · 2024-08-07
>
> We thank the reviewer for their feedback and questions. It appears that the primary concern lies in the fact that our results only hold for shallow networks and linearly separable data. We emphasize that these are limitations also present in the existing literature on benign overfitting in non-linear models / networks in the rich feature learning regime (i.e., not leveraging the NTK framework). The primary contribution of our work is to relax the relationship between the input dimension and the number of points: in particular past works require a very large input dimension, $d = \Omega(n^2 \log(n))$ or worse. In contrast, our results are for $d = \Omega(n)$ which is far more realistic. Furthermore to achieve this required new and highly different techniques to those used in the existing literature. As a result we would respectfully ask the reviewer to reconsider their rating. We proceed to address in more detail the specific weaknesses highlighted.
>
> > This paper mainly studies a specific shallow leaky ReLU network. The applicability of the results is limited to deeper neural networks or other complex models such as CNN and transformer.
>
> > The results rely on a specific linearly separable data distribution assumption.
>
> - Our setup involving a shallow ReLU network combined with the proposed data model is a) in-line with relevant and recent literature on the topic and b) represents perhaps the simplest and most natural setting involving a non-linear model in which one can study benign overfitting. Even in this simple setup many theoretical questions remain open: in particular, all past works require a near orthogonality condition on the data. As the emphasis of our work is to relax this technical condition we decided to work with this canonical setup. Deriving theory for more complicated settings, in particular deep networks and transformers, is clearly an important open problem. Such theory will almost certainly be built upon a solid understanding of the simple, shallow case.
>
> > The dependence of Theorem 3.2 and Theorem 3.3 on $n, d, k$ does not cover all cases. For other cases, what are the characteristics of overfitting?
>
> - In Theorems 3.2 and 3.3, the assumption $k = O(n)$ is minimal, since we always have $k \leq n$. The relaxation $d = \Omega(n)$ from $d = \Omega(n^2 \log n)$ is a main improvement of our results from prior work. We believe that the regime $d = O(n)$ will lead to different results than the ones observed in our paper, and will involve different techniques to study. In particular, if $md \leq n$, then the network is underparameterized and it is likely impossible to perfectly fit the data. When $d = 1$ and the network is sufficiently wide, there are some partial results in the literature which suggest that the network will perfectly fit the data, but the overfitting is not benign [1]. To the best of our knowledge the case $d = O(n)$ remains open.
>
> [1] Guy Kornowski, Gilad Yehudai, and Ohad Shamir. From tempered to benign overfitting in ReLU neural networks. *Advances in Neural Information Processing Systems*, 2023.

---

> > ### Comment · Reviewer_1LbK · 2024-08-12
> >
> > Thank you for your thoughtful response. However, I believe there may have been a misunderstanding of the author on my question concerning Theorems 3.2 and 3.3. Specifically, these theorems do not cover all possible cases of n and k. For certain values of α (such as 1/m), these do not provide a clear phase transition point/line. There are cases where the conditions of Theorems 3.2 and 3.3 are both not satisfied, so we can not know whether the overfitting is benign or non-benign, or whether there is a sharp transition between benign and non-benign. I believe it would be beneficial for the paper to include a discussion on this point. Additionally, I would like to request a more detailed explanation of the technical innovations and challenges presented in your work.

---

> > > ### Author Response · Authors · 2024-08-13
> > >
> > > Thanks for your response and clarification; we attempt to better address your questions below.
> > >
> > > - Theorem 3.3 is not a benign or non-benign overfitting result; it is just a matching lower bound to the upper bound of Corollary 3.2.1 valid in a regime largely overlapping with that of Corollary 3.2.1.  The non-benign overfitting result is Theorem 3.4.  We do agree these is a gap between Theorems 3.2 and 3.4, but the gap is in the parameter $\gamma$: Theorem 3.2 requires $\gamma = \Omega(1/k)$ while Theorem 3.4 requires $\gamma = O(\alpha^3/d)$.  The case where $\gamma$ lies between these bounds is interesting but we did not study it in this paper.  We remark that since we can allow $d = \Theta(n) = \Theta(k)$ (the regime where we improve upon existing benign overfitting results), it is possible that the lower bound and upper bound are tight up to a constant factor.  This is in contrast to previous work in benign versus non-benign overfitting [1], where the gap was $\gamma = \Omega(1/n)$ versus $\gamma = O(n^{-3/2})$.
> > >
> > > - When applying Theorem 3.2 to leaky ReLU networks, the dependence on $m$ cancels out; there is no reference to $m$ in Corollary 3.2.1 so the case $\alpha = 1/m$ is not special. The general question of small values of $\alpha$ is a legitimate issue; as $\alpha \to 0$, the activation function converges to ReLU, and our bounds get worse. In the context of leaky ReLU networks, we are interpreting $\alpha$ to be a fixed positive constant which does not vary with other parameters and is an attribute of the architecture.
> > >
> > > - We will add more details distinguishing the contributions of our work with prior works in future revisions.  The main technical contribution is Theorem 3.2, which bound the generalization error above for an approximately margin maximizing algorithm, including but not limited to gradient descent.  This result was challenging to show as it assumes only data assumptions and that the network achieves zero loss on the training data and has bounded norm.  Therefore, previous implicit bias results for gradient descent do not apply.  We were able to show this theorem using the fact that a sufficiently wide Gaussian random matrix with high probability is well-conditioned.  While this fact is known, its application to benign overfitting in neural networks is novel.  This allows us to (1) bound the norm of the max-margin classifier of the training data and (2) argue that if the network has small weights then it cannot fit the training data using only feature noise, so if it achieves zero loss it must have learned a strong signal.
> > >
> > > [1] Erin George, Michael Murray, William Swartworth, Deanna Needell. Training shallow ReLU networks on noisy data using hinge loss: when do we overfit and is it benign? *Advances in Neural Information Processing Systems*, 2023.

---

### Author Rebuttal · Authors · 2024-08-07

We thank the reviewers for their time and efforts in providing feedback on our work. Two actions we can take forward in future revisions are as follows.

- **Improving the presentation**: We will better connect Section 3, where the key results are presented, with the proof sketch given in Section 4. In doing so we can provide better intuition for the reader as to the role of the number of corruptions $k$ and the signal strength $\gamma$, in particular their role in the derived bounds. We will also better elucidate the differences between our work and that by Brutzkus et al. and George et al. in the related works section.
- **Including experiments**: We will include in future revisions a new section in the appendix containing numerics supporting our theoretical results. Preliminary plots are included in the attached pdf. We offer the following explanation of the two experiments.
    - Under Assumptions 1 and 2 a key contribution of our work is to show $d = \Omega(n)$ is sufficient for benign overfitting. Note, by contrast other works require $d = \Omega(n^2\log(n))$ or higher. To test this experimentally, using gradient descent and the hinge loss we trained the inner layer weights of shallow, two layer leaky ReLU networks, varying both the size of the training sample $n$ as well as the input dimension $d$. To estimate the generalization error, for each setting of $n$ and $d$ twenty trials were conducted. Within each trial a training sample was drawn from the data model (described in Definition 2.1), a network as per Assumption 2 was initialized and trained to zero hinge loss and the validation error computed on a validation sample. The generalization error across the trials was then averaged. The results of this experiment are provided in Figure 1.

    - We remark that the number of corruptions $k$ in our experiments is fixed at $10\%$ of $n$. Considering the figure, when $d =n$ we observe that the generalization error matches the corruption rate, suggesting proportional, tempered overfitting. We note that our results do not cover or explain this setting. However, for $d >n$  we observe what seems to be a decrease in the generalization error with $d$: indeed, for $d = 10 n$ versus $d = n$ we observe a ten-fold decrease in the generalization error, dropping from the corruption rate of $10\%$ to $1\%$ error. This suggests, inline with our theory, that the generalization error is a sharply decreasing function of $d/n$.

   - In the second experiment (Figure 2), we vary the number of data points $n$ and the signal strength $\gamma$, holding constant the ratio $d/n$, the corruption rate $k/n = 0.1$, and the hidden dimension $m = 64$. We plot the generalization error as a function of $\gamma$ over different values of $n$. We find that for all values of $n$, the generalization error decreases steeply at a certain threshold, then levels off at a fixed small value. For higher values of $n$, this drop-off of generalization error occurs sooner. This is in agreement with our theoretical results where we found that benign overfitting occurs at the threshold $\gamma = \Omega(1/k)$ (which is in this case $\Omega(1/n)$). We also see that the generalization error for large values of $\gamma$ is similar across different values of $n$. This effect is also predicted by Corollary 3.2.1, since we scale both $d$ and $k$ proportionally to $n$.

---

### Decision · Program_Chairs · 2024-09-25

**Decision:**

Accept (spotlight)

**Comment:**

In this paper, the authors establish conditions under which a two-layer leaky ReLU network (trained with hinge loss) provably exhibits benign or non-benign overfitting on noisy linearly separable data.  Previous works establish comparable results but require that the data to satisfy a near orthogonality property, requiring that the input dimension scales like n^2 log n, where n is the training sample size.  In the present work, the authors use an alternative proof strategy and only require that the input dimension scales like n.  The proof technique is based on an approximate margin maximization property, which is responsible for the improved results and may allow for additional results to be proved.  While the results are limited to the case of linearly separable data, these assumptions are standard in recent works in the area.  The strengths outweigh the weaknesses, and the paper is recommended for acceptance.